

# Horizontal geometry of trade-wind cumuli – aircraft observations from shortwave infrared imager versus radar profiler

Henning Dorff[1,2], Heike Konow[1,3], and Felix Ament[1,3]

[1]University of Hamburg, Hamburg, Germany
[2]International Max Planck Research School on Earth System Modelling, Max Planck Institute for Meteorology, Hamburg, Germany
[3]Max Planck Institute for Meteorology, Hamburg, Germany

**Correspondence:** Henning Dorff (henning.dorff@uni-hamburg.de)

**Abstract.** This study elaborates how aircraft-based horizontal geometries of trade-wind cumulus clouds differ whether a one-dimensional (1D) profiler or a two-dimensional (2D) imager is used. While nadir profiling devices are limited to 1D realisation of the cloud transect size with limited representativeness of horizontal cloud extension, 2D imagers enhance our perspectives by mapping the horizontal cloud field. Both require high-resolution to detect the lower end of the cloud size spectrum.

In this regard, the payload aboard the High Altitude and LOng Range Research Aircraft (HALO) achieves a comparison and also a synergy of both measurement systems. Using the NARVAL-II campaign, we combine HALO observations from a 35.2 GHz cloud and precipitation radar (1D) and from the hyperspectral 2D imager specMACS, having a 30 times higher along-track resolution and compare their cloud masks. We examine cloud size distributions in terms of sensitivity to sample size, resolution and the considered field of view (2D or 1D). This specifies impacts on horizontal cloud sizes derived from the

across-track perspective of the high-resolution imager in comparison to the radar curtain. We assess whether and how the trade-wind field amplifies uncertainties in cloud geometry observations along 1D transects through directional cloud elongation.

  Our findings reveal that each additional dimension, no matter of the device, causes a significant increase of observed clouds. The across-track field yields the highest increase in the cloud sample. The radar encounters difficulties to characterize the trade-wind cumuli size distribution. More than 60 % of clouds are subgrid scale for the radar. While the radar cannot resolve clouds

shorter than 200 m and has a lower sensitivity, the amount of small invisible clouds leads to deviations in the size distribution. Double power law characteristics in the imager based cloud size distribution do not occur in radar observations. Along-track measurements do not necessarily cover the predominant cloud extent and inferred geometries lack of representativeness. Trade-wind cumuli show horizontal patterns similar to ellipses with a mean aspect ratio of 3:2. Instead of circular estimations based on the 1D transect, elliptic fits maintain the cloud area size distribution. Increasing wind speed tends to stretch clouds more

and tilts them into the wind field, which makes transect measurements more representative along this axis.





# 1 Introduction

Marine shallow cumulus clouds exhibit the highest frequency of occurrence in the trade-wind region (Brueck et al., 2015; Bony et al., 2017) and affect climate up to global scale (Siebesma, 1998; Wing and Emanuel, 2014). Nevertheless, these clouds are limitedly understood (Vial et al., 2017; Nuijens and Siebesma, 2019). According to general circulation models (GCM), they drive the largest spread in cloud feedbacks (Medeiros et al., 2015; Ceppi et al., 2017). Not only trends in cloud cover, changing the Earth radiation budget (Rieck et al., 2012; Nuijens et al., 2015; Bony et al., 2015), but also horizontal radiative transport through the cloud edges are crucial (Zinner et al., 2006; Hinkelman et al., 2007). Nuijens and Siebesma (2019) review an underestimation in coupling of cloud properties and the boundary layer (BL) regarding cloud geometries, organisation, trade winds (e.g. Bretherton and Blossey, 2017; Tompkins and Semie, 2017; Stevens et al., 2019b) and precipitation (Vogel et al., 2016). To grow our knowledge of cloud controlling factors (CCFs) and their role in climate change (Siebesma et al., 2009; Klein et al., 2018), the macrophysics, i.e. cloud geometries have to be sampled together with predominant BL conditions.

Due to the omnipresence of trade-wind cumuli over the tropical oceans, their observation requires remote sensing. Vertical profilers, such as on CloudSat (Stephens et al., 2002), can measure overpassed clouds in vertical transects (Illingworth et al., 2007). However, since profilers characterise cloud properties in a one-dimensional (1D) way, moving space-/airborne platforms require long-distance paths so that they are of limited use for local cloud cover estimates (Astin et al., 2001). Fixed locations, e.g. supersites like the Barbados Cloud Observatory (BCO) furnish ground-based cloud profiling downstream the Trades (Stevens et al., 2016). They need several hours of cloud passage until cloud cover variance decreases (Klingebiel et al., 2021). Both 1D methods estimate cloud properties outside transects statistically or assumptions of circular cloud shapes (van de Poll et al., 2006; Romps and Vogelmann, 2017) which constrains the utility for model verification (Winker et al., 2017).

Vertically pointing cloud imagers observe clouds in the two-dimensional (2D) horizontal plane. In addition to enlarging the field of observation, they face the challenge that trade-wind cumuli show sizes ranging over several orders of magnitude (Zhao and Girolamo, 2007; Wood and Field, 2011; Mieslinger et al., 2019) as they can resolve properties of trade-wind cumuli down to decameter scale (Zhao and Girolamo, 2007). Clouds being subgrid-scale for common hectometer profiler resolution may have extensive unresolved radiative impacts or be misinterpreted by erroneous cloud size distributions and unresolved horizontal cloud properties. In a large spaceborne framework of shallow cloud properties, Mieslinger et al. (2019) bridge the gap between horizontal cloud geometries and trade-wind BL conditions from reanalyses. However, most spaceborne imagers are inadequate to detect sharp water-vapour gradients (Chazette et al., 2014) or to distinguish between overlapping clouds.

A synergy of along-track cloud profiling with capturing their 2D horizontal structure at high-resolution, while measuring BL conditions simultaneously, is barely provided. Yet, the payload of the High Altitude and LOng Range Research Aircraft (HALO) fulfils such requirements (Stevens et al., 2019a). The Next-Generation Aircraft Remote Sensing for Validation Studies (NARVAL) II was the first campaign over the Tropical Atlantic, where the 2D imager specMACS (Ewald et al., 2016) was mounted in a nadir perspective aboard HALO and complemented profiling observations based on radar and lidar. Therefore,





we elucidate capabilities of such modern airborne remote sensing in characterising horizontal geometries of shallow trade
wind cumuli. In this context, the imager resolution, which is high against the radar, receives detailed attention, as Large Eddy
Simulations (LESs) become capable to capture cloud properties at hectometer scale (Stevens et al., 2020). We investigate *what
remains unresolved in statistics of horizontal trade-wind cumulus geometries using radar profiler having hectometer along-
track resolution?* The emerging cloud size distributions that include decameter scale feed our modelling understanding.

Profilers restriction to 1D transects may lead to sampling uncertainties for airborne cloud statistics (e.g. Gutleben et al., 2019),
whereas the imager detects the horizontal 2D cloud field. We evaluate how airborne 2D imagers enhance our perspectives on
cloud geometries compared to 1D profiling. Similar to LES studies (Barron et al., 2020), which aim to transfer from 1D cloud
lengths to 2D horizontal sizes, we assess: *How representative are cloud sizes derived from 1D transects after considering a
nadir 2D field of view (FOV)? Which simplified shape approximates the horizontal cloud projections?* Since Mieslinger et al.
(2019) identified trade winds as the most powerful CCF, we further infer: *To what extent do strong trade winds and corre-
sponding cloud elongation play a role in the assessment of horizontal cloud sizes using transect measurements?*

Pursuing these four questions, our manuscript is structured as follows: Section 2 specifies the used cloud detection products
from HALO, namely the radar and imager cloud masks. Periods ensuring collocated observations are identified. Section 3
encompasses our applied synergy of both cloud masks in order to gain cloud geometries in the horizontal plane. Section 4
adresses the first two research questions, viewing on the suitability of radar transects to characterize horizontal trade-wind
cumuli size distributions. This section also searches for simplified shapes that approximate horizontal cloud projections. As
application, Section 5 is dedicated to wind-induced impacts on horizontal cloud elongation and their relevance when using
transect measurements. The conclusions look on abilities of a prospective flight campaign for precedent research questions.

## 2 Airborne observations of trade-wind cumulus clouds

We consider observations from the research aircraft HALO, which is characterised by a high cruising level and long range
(Krautstrunk and Giez, 2012). HALO is equipped with several remote sensing devices measuring in different spectral ranges
(Stevens et al., 2019a). Our comparison of trade-wind cumuli geometries from airborne profiler and imager concentrates on a
1D profiling cloud and precipitation radar and a hyperspectral imager enabling the determination of horizontal cloud geometries
in 2Ds. This section depicts both devices and their cloud masks being used. The periods of collocated observations are given.

### 2.1 Cloud and precipitation radar

Aboard HALO, the cloud and precipitation radar MIRA-35 (manufactured by METEK GmbH) is mounted in a downward
pointing configuration. MIRA-35 is a monostatic, pulsed and magnetron Ka-band Doppler radar operating in 35.2 GHz within
the water-vapour window (Ewald et al., 2019). The output in a sampling rate of 1 Hz mainly comprises radar reflectivity,
Doppler velocity, spectral width and linear depolarisation ratio. Raw measurements are provided in a vertical resolution of
28.8 m. For typical cruising levels at 13 km and a typical aircraft speed around 200 m/s, the radar sensitivity is about -30 dBZ,
whereby footprint size at the surface is about 130 m. By infiltrating clouds, the profiling radar enable the observation of vertical



cloud structures and represents a fundamental component of airborne cloud analysis. Clouds are resolved at roughly 200 m in the along-track. Together with the HALO Microwave Package (Mech et al., 2014), the processed data is provided in a unified grid with a vertical resolution of 30 m and a time resolution of 1 Hz (Konow et al., 2018, Konow et al., 2019).

## 2.2 Hyperspectral imager

90  The hyperspectral imager specMACS (Ewald et al., 2016) was developed at the Meteorological Institute of the Ludwig Maximilians University (LMU) Munich, with the goal to detect clouds and aerosol optical properties, as well as aerosol tracers. This is done using two passive hyperspectral cameras which measure the incoming solar radiation within the 400–2500 nm wavelength spectrum. One camera measures radiances in the visible and near infrared spectrum (VNIR) (400–1000 nm). The second camera measures within the shortwave infrared (SWIR) spectrum in the range of 1000–2500 nm (Ewald et al., 2016).

95  Both cameras are line-scanning slit spectrometers that capture radiances in two dimensions. When the aircraft overpasses a certain domain, the downward pointing cameras look on clouds in the along-track and across-track perspective from above. This results in a 2D observation field. For the purpose of this study, we focus on the cloud products gained from the SWIR camera. This camera measures incoming radiation in 320 across-track pixels and in 256 wavelengths and enable the observation of the across-track cloud top in a FOV of 35.5°.

## 100  2.3 Cloud masks

For the cloud geometry analysis, the cloud masks from radar (Sec. 2.1) and imager (Sec. 2.2) are prerequisite. Their different measurement principles pronounce in individual cloud mask specifications (Table 1). The radar cloud mask is also provided on the unified grid (Sec. 2.1) and derived from the measured radar reflectivity following the methods of the BCO radar cloud mask algorithm (Konow, 2020). Since reflectivity values were already filtered, i.e. adjusted for constant background noise, the

105  remaining detected signals represent clouds and precipitation. As soon as a reflectivity value is recorded, the pixel is assigned as cloud (value of 1), otherwise clear. Measurement gaps were interpolated and the clouds were morphologically closed. Clouds being measured very close to each other (1 s which represents around 200 m) are connected.

To identify clouds from the imager, the intuitive approach of brightness thresholds does not yield unambiguous results as various processes affect the radiance values of each pixel. These are mainly the brightness dependency on the Sun's position,

110  atmospheric water vapour and the presence of sun glint, meaning the reflection of sunlight at the ocean surface under specific conditions. Sunglint-contaminated pixels can have higher radiance values than cloud pixels. Hence, the imager cloud mask is a best-estimate of a water-vapour adapted brightness mask together with a simulated sun glint mask based on L. Tsang (1985). In contrast to the 1 Hz radar cloud mask, the imager cloud mask has an along-track resolution based on a frame rate of 30 Hz. This cloud mask was developed by Gödde (2018) and is accessible on the *macsServer* (Kölling, 2020).

115  Besides the along-track axis, both cloud masks provide a second dimension. While the radar mask considers the vertical profile in a 30 m resolution, the imager mask represents the 2D horizontal cloud field seen from above. For a typical cruising altitude of 10 km, the FOV of specMACS corresponds to an across-track viewing distance of about 6.4 km. With a frame rate of 30 Hz, the spatial resolution of the imager cloud mask results in roughly 18 m along and 38 m across the flight track at this altitude.



**Table 1.** Cloud mask specifications from radar and imager specMACS (SWIR camera)

| Specifications | cloud and precipitation radar | specMACS SWIR camera |
|---|---|---|
| major instrument characteristics | Ka-band Doppler radar at 35.2 GHz | hyperspectral camera measuring from 1000–2500 nm in 256 spectral channels |
| threshold variables for cloud masking | radar reflectivity factor | brightness (1000–1900 nm) considering sun position, water vapour, sunglint |
| along-track resolution | unified grid in 1 Hz | frame rate of 30 Hz |
| observation field from second axis (y or z) | vertical (z): from aircraft to ground in a resolution of 30 m | across-track (y): FOV 35.5° resolved with 320 spatial pixels; typical viewing distance of 5 to 8 km from cruising level |

## 2.4 Synchronous cloud observations during NARVAL-II

To compare horizontal trade-wind cumuli geometries from airborne profiler and imager, we use observations from the NARVAL-II campaign, conducted in August 2016 at the edge of the Intertropical Convergence Zone (ITCZ) eastern of Barbados under the influence of the North Atlantic trade winds (Stevens et al., 2019a). For that, overlapping measurements from radar and imager are indispensable. Ten research flights (RFs) were performed in shallow and deep convection (Konow et al., 2019). However, we had to exclude three flights without reliable radar operation and the first transfer flight (RF01), as measurements were mostly taken north of the tropics ($> 20°$N). The remaining flights suffer some interruptions in reliable overlapping cloud observations (Fig. 1) due to window freezing in front of the imager at high cruising levels that prevented reliable cloud detection and other occurring disturbances due to issues at the radar and its calibration (Ewald et al., 2019).

Around half of the duration of the remaining RFs provide suitable, collocated measurements. In total, this corresponds to more than 24 flight hours. Circle flight patterns were mainly performed to examine large-scale divergence in Bony and Stevens (2019), wherein dropsondes (Busen, 2012) captured the vertical atmospheric state. We make use of these sonde profiles in Sec. 5 to characterize the trade-wind field in the vicinity of shallow convection and to relate the winds to the geometries of embedded cumulus clouds. This analysis substantially benefits from frequent sonde releases during RF03 and RF06 (Fig. 1).

## 3 Derivation of cloud geometries

While Gutleben et al. (2019) examined lidar-based cloud properties from NARVAL-II under influence of Saharan dust, this study complements the statistics more generally under aspects of differences in radar profiler and imager perspective and their resolution. For the derivation of single cloud geometries and inter device comparison, the different viewing directions of both cloud masks have to be contemplated. While 1D along-track cloud geometries can be inferred immediately from the time axis, when considering the cruising speed, the imager across-track axis is initially given in camera pixels and has to be transformed to distance based coordinates if 2D geometries are desired. The major steps for both requirements are specified.





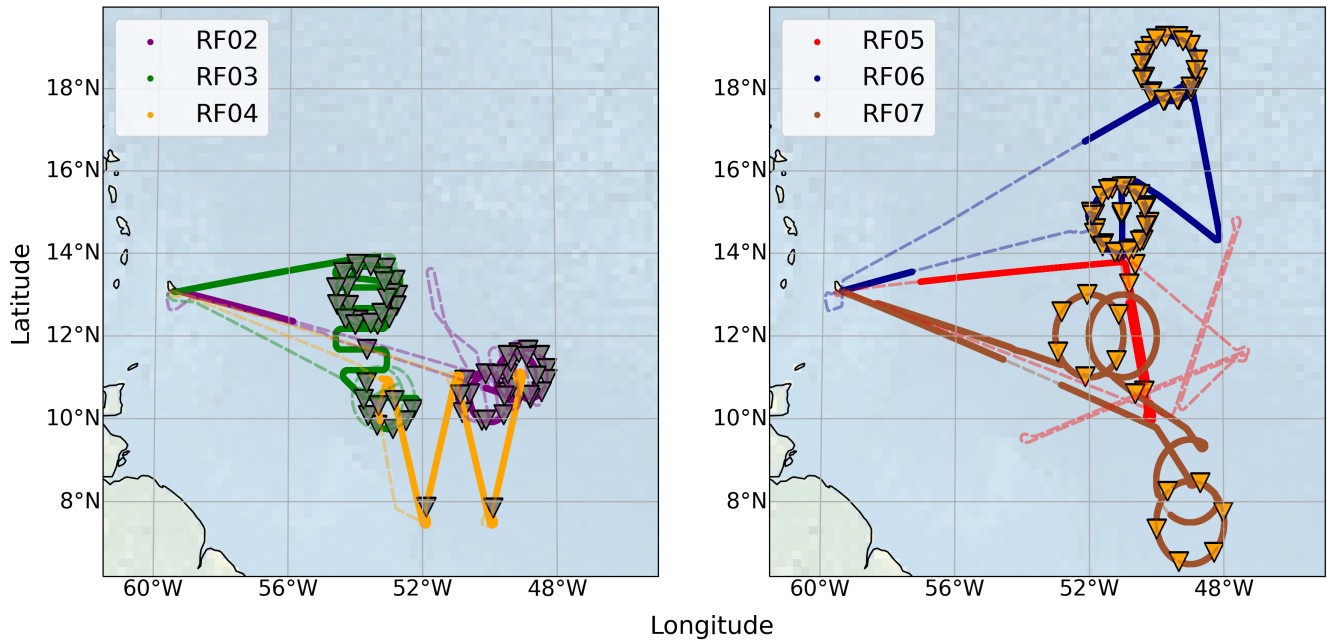

**Figure 1.** Considered flight tracks from NARVAL-II (dashed lines) that provide observations of tropical clouds from the radar and imager specMACS. Periods in which both devices (radar and imager) were operating reliably and simultaneously are highlighted as bold lines. Dropsonde releases less than 15 minutes away from overlapping measurements of radar and imager are indicated as triangles. Background map made with Natural Earth.

## 3.1 Viewing direction and cloud object identification

Due to the alignment, the imager FOV is 2.6° ahead of the radar, so that both devices look at the same cloud at different time frames. This time shift depends on the cloud top height (CTH); The lower and further to the aircraft the clouds are, the more the detection times deviate between the devices. This leads to non-negligible time offsets for cloud detection. Following the methods described in Höppler et al. (2020), the offsets are calculated using the CTH, which we obtain from the radar. The offsets are determined from the aircraft distance to the clouds and the aircraft speed, and subsequently applied to the radar cloud mask. Hence, same clouds are referred to equivalent time frames in both devices.

Single clouds are identified as coherent objects by a connected component analysis, where each cloud pixel is connected with its neighbour cloud pixel (Fig. 2). According to the eight-connectivity algorithm (Rosenfeld and Kak, 1982), we consider neighbouring cloud pixels as coherent when they are connected at one of their edges or corners. Each cloud is assigned an identification number (ID). By comparing the radar reflectivity with the cloud IDs (Fig. 2a,b), the morphological closing applied to the cloud mask becomes visible. For the along-track axis during the given scene, it is already notable that the imager (Fig. 2c,d) counts many more clouds than the radar. This worthwhile remark is discussed in more detail in Sec. 4.

**Figure 2.** Cloud scene during RF02 from 17:28:30 to 17:30:15 Coordinated Universal Time (UTC), vertically zoomed in from the radar (a,b) and as seen from specMACS (c,d). For each device, the upper panel shows measurements of the cloud scene, namely the reflectivity factor in dBZ (radar) and radiances given in mW m$^{-2}$sr$^{-1}$nm$^{-1}$ of the 1600 nm channel from the SWIR camera (imager specMACS). The yellow curtain in c) represents the radar FOV. Each lower panel indicates the number of clouds based on their IDs obtained from the connected component analysis applied to the cloud masks of radar (b), and specMACS (d) respectively.





## 3.2 Distance based pixels

To characterize 2D cloud geometries in the horizontal projected plane, each cloud object has to be transferred into a local distance based coordinate system (Fig. 3). For that, the transformation takes into account the coordinate frame of the SWIR camera $C_{\mathrm{SWIR}}$, where across-track pixels have fixed viewing directions in reference to HALO. Since the geometric field covered by a single imager pixel depends on the flight attitude, the initial coordinate frame $C_{\mathrm{SWIR}}$ is aligned differently at each time frame. To transform the direction vectors from $C_{\mathrm{SWIR}}$ into the target coordinate system, rotation matrices, that include the flight attitude (i.e. pitch, yaw and roll angle), yield the pixel viewing directions in the local reference system providing both axes in meters. Dividing the vertical distance between the camera at cruising level $z_{\mathrm{Camera}}$ and radar-based CTH, given with an accuracy of 30 m, results in the slanted distance $d$ that specifies the stretching factor for the vector projection down to the CTH layer $z_{\mathrm{CTH}}$. Tracking downwards along $d$, cloud pixels are referenced to meters. This projection causes cloud sizes to primarily depend on the distance to the CTH. When analysing 2D cloud geometries, following aspects need to be considered:

– The discrete along-track CTH values are interpolated in time and synchronized to the imager resolution to achieve a continuous along-track representation of CTH.

– When observing very thin low-level clouds from very high cruising levels above 10 km, due to too low sensitivity, the radar partly tend to underestimate CTH or in some cases does not detect a cloud in contrast to the imager in the overlapping FOV. Arising too low CTH can affect the interpolation and lead to erroneous continuous CTH representation. Therefore, a quality check includes the lifting condensation level (LCL), calculated and interpolated in time from the dropsonde profiles, as a threshold approximation for minimum CTH. CTH values below the LCL are replaced.

– The across-track cloud observation by the imager requires CTH estimates also exterior of the radar transect. As a simplified assumption, the derived CTH is set constant in the across-track direction.

– In order to mitigate the projection issues for across-track clouds, CTH uncertainties are dammed by only considering shallow convection regimes with lower CTH variability. This restriction is also advantageous regarding sources of errors stemming from the imager viewing geometry: The further clouds are located at the edges of the FOV, the larger the pixel viewing zenith angle and the more clouds are actually detected from the side. In contrast, if clouds are shallower and lower, they are less observed from the side (Henderson-Sellers and McGuffie, 1990).

– As uncertainties of CTH, mainly in the across-track direction, prevail in a certain amount even in shallow convection, we use the along-track variability of CTH as an estimate of spatial CTH uncertainty to be included in the projection.

– Periods that the radar identifies with a high amount of multi-level and vertically overlapping clouds are excluded, as the imager cloud mask does not account for different height levels and merges these clouds.

– Clouds extending out of the FOV are neglected as their geometries exterior of the FOV are not obtainable. In addition, clouds having an along-track cloud size bigger than the imager across-track FOV are excluded, as otherwise large clouds orientated along the flight course would be considered preferentially and cloud size statistics may be biased.





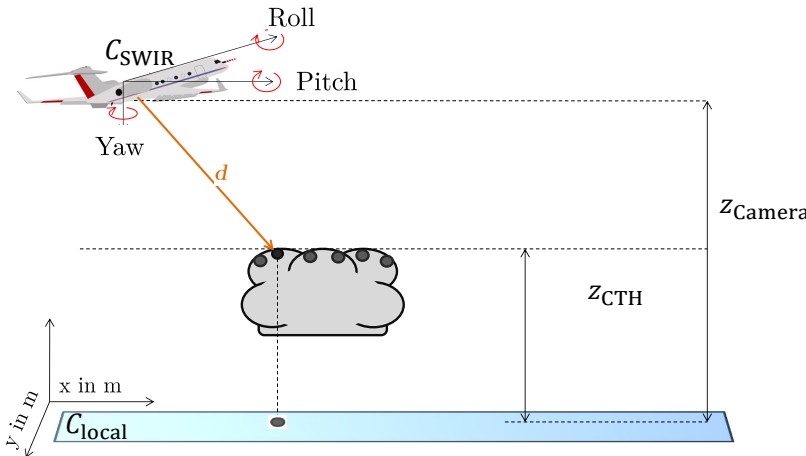

**Figure 3.** Illustration of the transformation from cloud pixels in the camera coordinate system $C_{\mathrm{SWIR}}$ into the local distance based reference system. Aircraft yaw, pitch and roll axes are defined in the top left-hand corner.

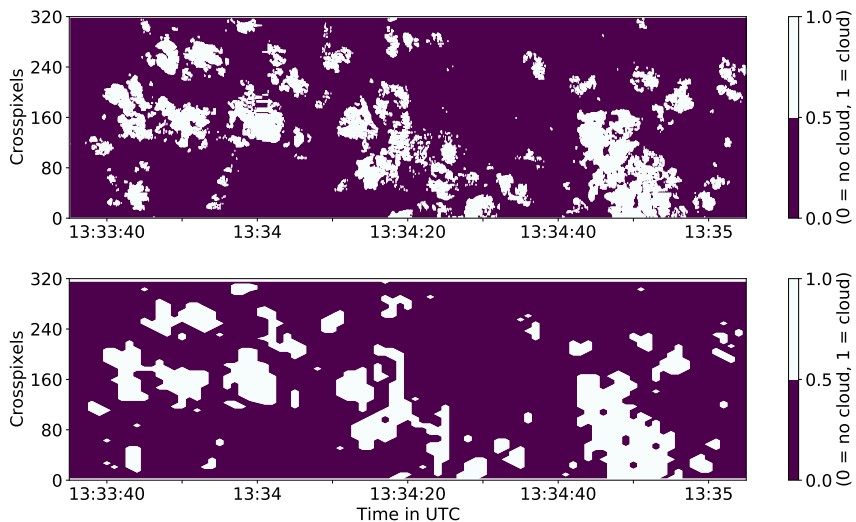

**Figure 4.** Imager cloud mask (specMACS) of shallow cumuli from RF07 in standard (top) and coarse-regridded resolution (bottom).

### 3.3 Regridded imager cloud mask

To specify resolution impacts on cloud geometry statistics separately from the impact of the FOV (1D and 2D), a coarse-gridded imager cloud mask mimics the spatio-temporal radar resolution having a time-resolution of 1 Hz and a larger footprint of each pixel, while still considering the across-track FOV. Resulting mean cloud cover below 0.5 inside coarse gridded pixels are labelled as clear, and vice versa if above. The emerging cloud mask is shown in Fig. 4 for an exemplary scene. This coarsening constitutes a conceptional simplification, as the sensitivities are by no means trivial and idealized in this regridded mask.



### 3.4 Cloud size distribution

Among cloud geometries, cloud size distributions are of fundamental interest. The vast majority of trade-wind cumuli are rather small while the size distribution tails off towards rare larger clouds. Covering decameters up to several ten kilometres,
their size distributions are conventionally analyzed in logarithmic mode. Satellite observations found that trade-wind cumuli size distributions obey a power law distribution (e.g. Sengupta et al., 1990; Benner and Curry, 1998; Zhao and Girolamo, 2007; Mieslinger et al., 2019). The frequency $n(D)$ of the cloud size $D$, being the area-equivalent diameter, is described as:

$$n(D) = \alpha \cdot D^{\beta}, \tag{1}$$

with the slope $\beta < 0$ and $\alpha$ being a parameter. In logarithmic mode, $\beta$ can be obtained by a linear least-square fit. Due to
strong decay in frequency with increasing cloud size, Mieslinger et al. (2019) recommend to adopt exponentially increasing bin widths, designated as logarithmic bins. The transformation to logarithmic bins $n(\log(D))$ increases the slope value by one:

$$n(\log(D)) \propto D^{\beta+1}. \tag{2}$$

Analysing the distribution with logarithmic binning, the fitted cloud size distribution becomes less sensitive to bin size. Expo-
nentially increasing bin widths allow the full range of the distribution to be considered. The size bins of cloud frequency are equidistantly displayed and the underlying area thus represents the integral over the probability density. Also cloud area size distributions can be approximated by (double) power law distributions (Mieslinger et al., 2019). The estimation of 2D cloud sizes from 1D chord measurements as done for nadir space-/ airborne profiler is, however, challenging and a current state of research. Simplified cloud shapes facilitate statistical estimates for the 2D size (Romps and Vogelmann, 2017).
Several studies identified double power-law characteristics as the frequency decay intensifies abruptly above a certain cloud size. The interception of both fits is referred to as scale break, e.g. in Sengupta et al. (1990) and Zhao and Girolamo (2007). Mechanisms affecting its location are broadly discussed in current research. Some scientist claim that scale breaks are related to physical phenomena such as dynamical processes from interactions with the BL (Sengupta et al., 1990; Mieslinger et al.,2019). Others argue that scale breaks originate artificially resulting from undersampling (Heus and Seifert, 2013).
The desired framework allows the statistical derivation of relevant cloud geometries. Using the radar and imager cloud masks, we compare 1D along-track cloud sizes with the 2D cloud extension and assess impacts of 2D airborne observations. This comparison is based on impacts on the overall cloud size distribution in logarithmic binning according to Eq. 2. Applying logarithmic binning, we elucidate the presence of a scale break from airborne observations during NARVAL-II. First, we compare the along-track cloud sizes gained from the radar with those from the 2D horizontal imager FOV given at higher resolution.
Secondly, we elaborate the representativeness of these lengths for overall cloud area statistics. Horizontal cloud areas are inferred from the 2D imager cloud mask by referring to the distance based local coordinate frame (Sec. 3.2). Applying the Shoelace formula (Braden, 1986), we calculate cloud areas via a polygonal approach. Moreover, we reproduce the cloud area distribution by simplified cloud shapes.



## 4   Implications of resolution and of 2D against 1D observations on horizontal cloud geometry

Apart from dependencies of cloud representation on sensor sensitivities, varying spatial resolutions affect cloud masks by the contribution of small clouds (Zhao and Girolamo, 2006; Koren et al., 2008 and Fig. 4). As found in Stevens et al. (2019a), this pronounces on along-track cloud fraction aboard HALO. Differing viewing perspectives (1D, 2D) from radar and imager can also influence statistics of horizontal cloud geometries. Using the methods of Sec. 3, this section examines what remains unresolved in horizontal cloud size distributions when using airborne profiler having hectometer resolution. Regarding the

robustness of cloud geometries from 1D transects, this section investigates to what extent cloud geometries are reliable after considering a nadir 2D FOV. We aim for simplified shapes that can be attributed to horizontal 2D patterns of trade-wind cumuli.

### 4.1   Cloud sample size

One general drawback of airborne cloud geometry statistics is their limited number of clouds included. The sample size is affected by the cloud mask resolution and FOV. As seen in Fig. 4, regridding the imager mask to radar resolution enlarges

clouds by incorporating small holes and nearby clouds. Small distinct clouds decrease in their size or even disappear. The scene is thus represented with less clouds and may cause different cloud size spectra. When restricted to the overlapping 1D transect of radar and imager, it is important to note that cloud objects are counted differently due to lack of knowledge from the across-track perspective. Only a fraction of ID labelled clouds from the 2D field reaches into the radar curtain (Fig. 5). Otherwise, several cloud fragments, albeit belonging to one cloud, can appear distinctively within the curtain. When neglecting the across-

track, such fragments lead to a considerably different sample (higher amount of clouds) and cloud size misrepresentation.
During our synergy period (Fig. 1), strongly different cloud sample sizes between imager and radar confirm the complexity in unambiguous horizontal cloud representation (Fig. 5). Not surprisingly, the across-track perspective, which includes clouds invisible for the nadir radar curtain, leads to a considerably larger sample. The 2D imager gathered roughly 80,000 clouds, whereas the radar only around 1,000 over the course of the campaign. Within the overlapping 1D transect, the imager includes

less than one twelfth of all clouds from its 2D field. However, not only the 2D FOV but also the high along-track resolution of the imager cause a gain of more than 78000 clouds (98 %). When considering the coarse-gridded 2D imager mask presented in Fig. 4, the number of cloud objects decreases by around 80 % for all flights. Within the 1D transect, regridding induces a decline of 65 %. Nonetheless, the number of observed clouds therein remains twice as high as from the radar when distinct cloud fragments are considered as new clouds. This results from the fact that very small and very low clouds remain undetected

for several periods due to the lower radar sensitivity. In turn, the imager lacks vertical information. To estimate its impact, the radar cloud mask is projected horizontally by declaring time frames with at least one cloudy pixel in the vertical column as overall cloudy. This vertical lack of information results in a cloud number decrease of about 33 %.
We emphasize that given percentages can solely serve as benchmarks for the robustness of cloud sampling against missing spatial dimensions and resolution. Evidently, the different considerations of coherent cloud fragments translate into different

geometry statistics. This highlights the importance of multi-dimensional cloud observation and instrument redundancy. Overall, the increase in cloud number due to the 2D field outweighs that caused by the increase in resolution.





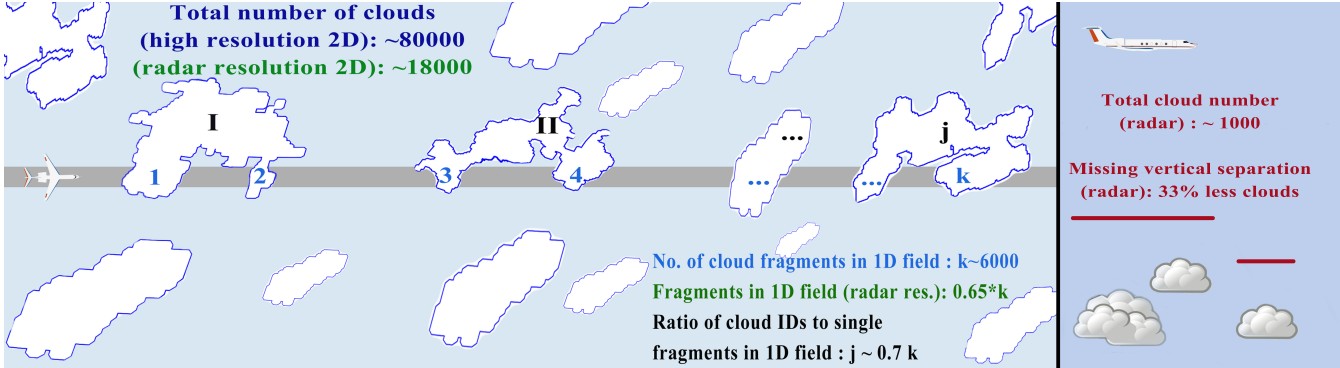

**Figure 5.** Quantitative impact on cloud counting during NARVAL-II when including the second dimension in the across-track (left) and vertical (right) for imager and radar. Left: Roman numbers exemplary represent ID-labelled clouds reaching into the radar curtain (grey swath not true to scale). Arabic numbers (in clouds) represent distinct cloud fragments reaching into the radar FOV. Right: Vertical distinction between overlapping clouds shows similar increases. Numbers outside clouds refer to the entire synergy period (specified in Sec. 3).

## 4.2 Impact of resolution on along-track cloud length statistics

Along-track lengths $L$ represent the horizontal cloud sizes that can be calculated from both nadir devices directly. For both NARVAL campaigns, Gutleben et al. (2019) provide lidar based statistics of cloud along-track lengths underneath HALO. The

FOV and resolution of the lidar is thereby comparable to the radar, however not its sensitivity. The fact that the radar detects less clouds than the imager specMACS within the matching FOV (Sec. 4.1) motivates further investigation of the cloud size statistics. For this purpose, the first assessment of impacts from collocated imager observations focusses on the intercomparison of cloud size distributions within their matching 1D FOV. This includes the regridded imager mask that mimics the radar resolution. For the relevant flight periods (Fig. 1), we can infer the robustness of the profiling curtain measurements by including

cloud-chord length distributions along additional parallel transects from the 2D field.

Within the overlapping FOV, the relative frequency of clouds undergoes a robust decrease with increasing cloud size for chord lengths above 500 m, no matter the resolution or device (Fig. 6). In detail, the frequency distributions deviate substantially between the devices and methods. For the cases in 1 Hz resolution (radar and regridded imager), clouds smaller than 300 m become sub-scale due to the flight speed. Above this scale, the regridded imager and radar show similar power-law distributions

for cloud lengths. According to the slope parameter $\beta$ (Eq. 1), the radar tends to a higher relative contribution of larger clouds in the along-track direction, leading to a weaker decay of the distribution. These deviations primarily result from different sensitivities of both devices. In particular, small trade-wind clouds occur within the lowest levels and are rather thin. Although such thin clouds can be long enough to be resolved, in truth, the radar does not catch them when distances between aircraft and clouds are quite far. Another minor reason for different slopes between radar and regridded imager is the ability of the radar

to separate clouds vertically, whereas the imager cannot distinguish between cloudy pixels from different heights and projects overlapping clouds onto a single cloud. This contrarily affects the distributions, as the imager overestimates the size of these





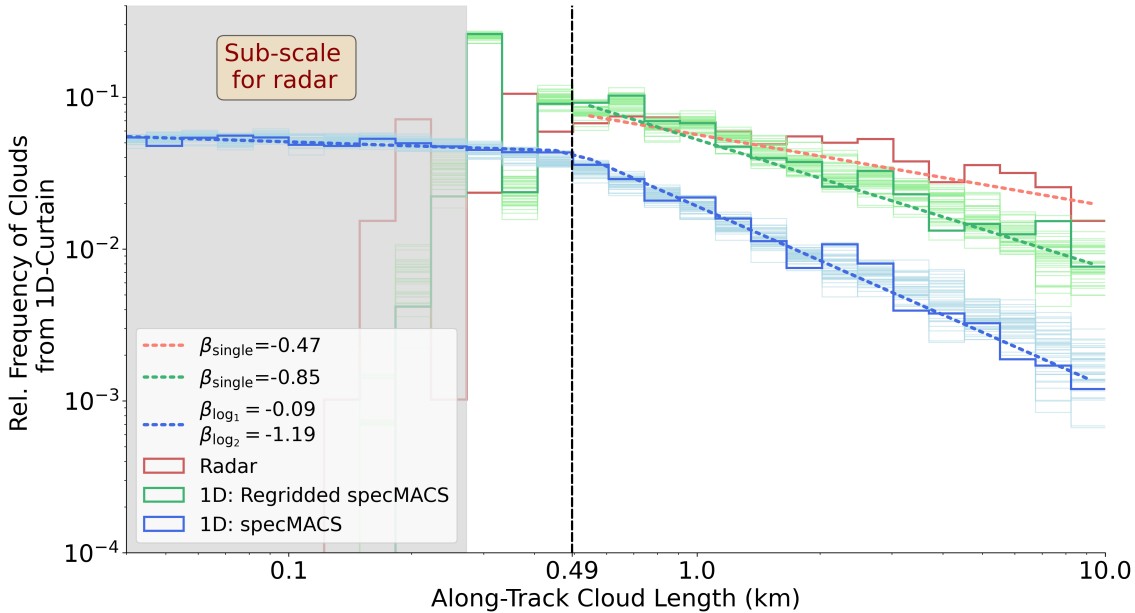

**Figure 6.** Along-track cloud size distribution from radar and imager (specMACS) within the 1D transect of their unanimous FOV (bold). For the imager, the high-resolution and regridded cloud mask (imitating radar 1 Hz resolution) are considered. Power-law fits are shown as dashed lines. The vertical dashed line represents the scale break resulting from the double power-law fit for the imager (blue distribution). Thin distributions illustrate cloud along-track chord lengths from successive curtains parallel to radar transect (discussed in Sec. 4.3.

clouds. Using the imager high-resolution, the lower part of the cloud size spectrum can be resolved. While for $L > 500$ m the slopes vary less in between both imager resolutions than in the inter-device comparison, the slope abruptly changes for cloud sizes below 500 m. The distribution thus can be described by a double-power law. As determined by least residuals (Sengupta et al., 1990), the slopes of the size distributions have values of $\beta_{\log_1} = -0.09$ and $\beta_{\log_2} = -1.19$ for smaller and larger clouds, respectively. Both fits yield an interception at around 490 m. This scale break is located close to radar resolution around 300 m.

### 4.3 Impact of the 2D FOV on along-track cloud size statistics

From now on, we extend our perspective on horizontal cloud geometries by adding the second (across-track) dimension. For instance, the relative frequency of small clouds, seen to be rather uniform in Fig. 6, is a caveat of the along-track length determination from the 1D curtain. Due to the fractal structure of clouds it is likely that they are overpassed at one of their edges, resulting in small chord length determinations. Referred to the statistical study of Barron et al. (2020), measured chord lengths below 100 m yield little to no information about the actual cloud size distribution, as they increasingly arise from the probability of touching cloud edges. If clouds were randomly distributed in the 2D FOV, relative frequencies of their 1D chord lengths should ideally yield an equivalent distribution regardless the curtain chosen. From the thin distributions in Fig. 6, we



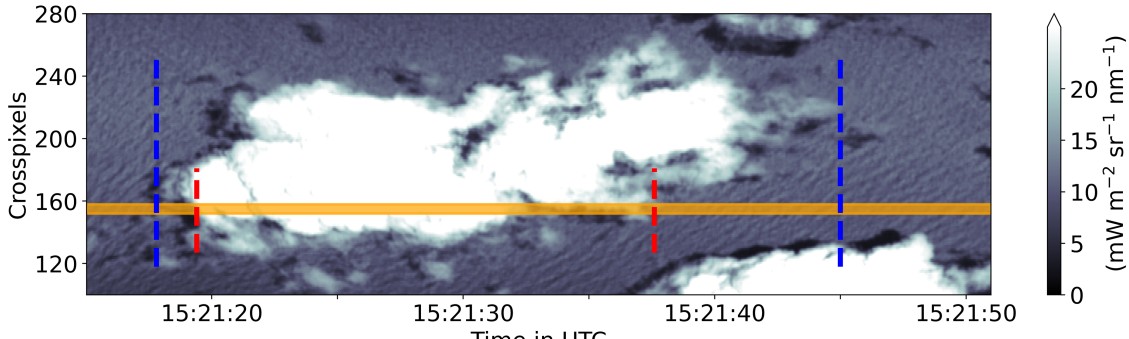

**Figure 7.** Cloud partly reaching into the FOV of the radar (orange swath), measured by the imager 1600 nm channel. Dashed lines indicate the along-track cloud length within the radar FOV (red) and maximum along-track cloud length for the entire across-track extension (blue).

note that all distribution characteristics occur more or less equivalently along other curtains parallel to the overlapping transect. Although the imager resolves decameter-scale clouds, cloud chord lengths from 1D transects merely serve as a rough guess for the distribution due to the contribution of cloud edges shifting the curves significantly towards small chord lengths (also in Fig. 2). Recorded cloud lengths can differ significantly from the actual cloud extension.

With regard to the along-track coordinate frame, the 2D consideration can reveal another representation of the along-track cloud size including the across-track perspective, which is sketched for an exemplary cloud in Fig. 7. The across-track perspective enables determining the maximum along-track cloud length. Although this length is an arguable size representation due to cloud's fractal structure, it quantifies the magnitude of the maximum error when concluding from 1D cloud chord lengths of a curtain to its overall statistics and thereby decrease the erroneous representation of seemingly small chord lengths from

actual cloud edges (Fig. 6). Moreover, this approach includes additional cloud objects from the across-track perspective that do not reach into the radar curtain, substantially increasing cloud sample size (Sec. 4.1). This maximum cloud size is also used by Romps and Vogelmann (2017) to approximate clouds as squares to transfer from 1D chord lengths to actual 2D sizes. Therefore, we compare the imager-based size distribution from the 1D transect in Fig. 6 with the maximum along-track cloud size distribution including all clouds from the 2D FOV at high-resolution (Fig. 8). Since twelve times the number of clouds

than in the 1D curtain are involved (Sec. 4.1), the representativeness of size distributions is more ensured anyway.

Concerning maximum along-track lengths, cloud sizes below 300 m, being sub-scale for radar resolution, represent more than 60 % of all overpassed clouds from the imager. We still observe the double power-law distribution in the 2D based sample. In addition to the resolution impact on the slope parameter $\beta$, Fig. 8 reveals that $\beta$ is very sensitive to the second dimension when using the entire FOV, which is essential regarding studies of cloud lengths obtained from 1D transects. The decay of the

distribution intensifies in both size regimes. Since the across-track consideration allows the distinction between small clouds and cloud edges leading to small along-track sizes, the representation of small clouds improves by quantifying the maximum along-track length as worst-case error. Intriguingly, decameter-scale clouds occur more frequently, whereas small medium-size clouds (L below 500 m) decrease in frequency. Clouds exterior of the radar FOV increase the sample non-uniformly. When





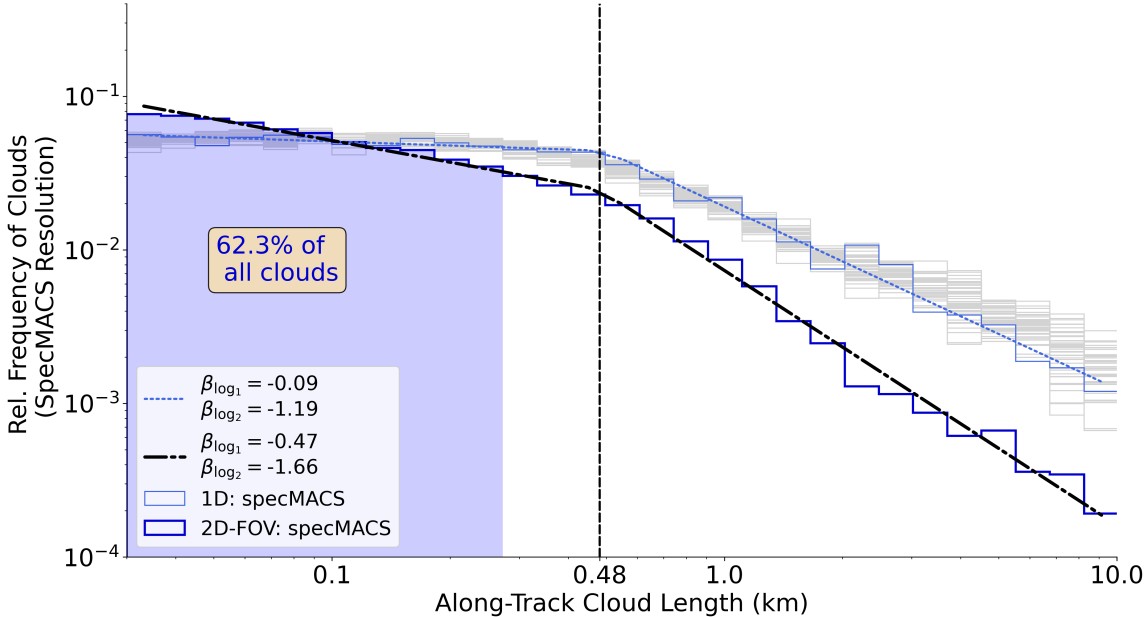

**Figure 8.** Comparison of along-track cloud size distribution from the imager (specMACS) based on 1D transects and the 2D horizontal field. Light blue and thin grey curves are identical to the 1D-based distributions in Fig. 6. The dark distribution and fitted curve represent the distribution for the maximum along-track cloud lengths based on the entire 2D FOV and the vertical-dashed line its scale-break. Shaded domains specify cloud sizes, being unresolvable for the radar at typical flight speed.

clouds are larger in the along-track axis, they generally tend to be larger in the across-track as well, often reach to across-track

edges covering the entire FOV and are embedded in a cluster of numerous small clouds around (Fig. 9). For measurements constraint on a narrow transect, this pronounces in a lower chance of small clouds around to be captured, whereas the largest clouds are captured anyway, regardless the resolution. Transferring from 2D to 1D transects, this flattens the distribution. Using the high-resolution imager, only very slight shifts of the scale break between the 2D and 1D observation field manifest. Both, the scale-break as well as the slope parameter, correspond well to literature values, e.g. Zhao and Girolamo (2007) and

Mieslinger et al. (2019). Thus, observations from the imager specMACS during NARVAL-II generally well represent typical trade-wind cloud size distributions. The radar resolution, however, is inadequate to resolve scale-break characteristics.

### 4.4   2D Cloud Size Statistics

Until this point, when identifying added values from the 2D imager, cloud sizes were still referred to their 1D along-track axis. However, it is the cloud covered area that can only be detected by the imager and that is locally relevant for the radiation bud-

get of the Earth (Bony et al., 2015; Brueck et al., 2015). Hence, we put now emphasis on the misrepresentation of along-track lengths for the effective 2D cloud size. Its 1D size projection, i.e. the area equivalent diameter $D$ can considerably deviate from along-track sizes (e.g. having a value between both distances for the cloud in Fig. 7). While nadir profilers are restricted



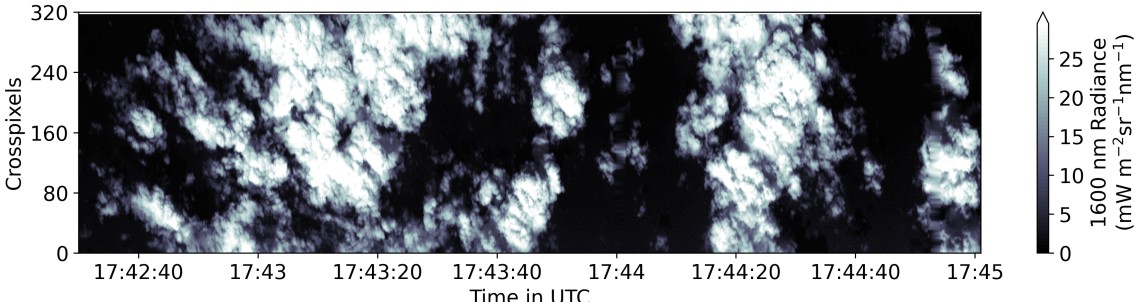

**Figure 9.** Cloud scene containing two clouds largely extending over the entire imager FOV. Radiances from imager 1600 nm channel.

to determine along-track lengths and need to estimate cloud-covered area from circular or statistical concepts, the 2D imager allows, after the coordinate-transformation (Sec. 3.2), the determination of horizontal sizes in all directions and the cloud

area thereof (Sec. 3.4). Significant amounts of directional orientated clouds may become crucial for the representativeness of profiler-based along-track size distributions. Elliptic shapes can enlighten dominant extension and orientation of clouds which are also proven to be a typical shape at the BCO (Stevens et al., 2016; Klingebiel et al.,2021). Therefore, an elliptic fitting algorithm performed in a least-square sense (Fitzgibbon and Fisher, 1996) is applied to each fully-detected cloud. To evaluate such shape simplifications and to specify limits of along-track cloud sizes, Fig. 10 depicts the cloud area distribution of the

polygonal (Sec. 3.4) and elliptic approaches and compares them to circular clouds assumed from the maximum along-track extension (as in Fig. 7). From this, we classify the polygons as our reference. The statistics confine oneself to shallow cumulus scenarios (CTH< 3 km) to limit projection errors due to CTH uncertainties (Sec. 3.2).

Similar to the precedent 1D size distributions based on the 2D FOV, cloud area distributions follow double-power laws (Fig. 10). Although small clouds, being sub-grid scale for the radar ($L < 200$ m), represent 80 % of all shallow clouds, they contribute

to less than 7 % to the total cloud-covered area. In terms of radiation budget, the miss of most of the clouds in radar resolution is thus less severe. The polygonal and elliptic approach reveal similar distributions. The observed trade-wind cumuli have a mean aspect ratio of roughly $3 : 2$. For each elliptic fit, we calculate the eccentricity that quantifies the elongation of clouds in an inter-comparable way. The cloud aspect ratio manifests in a mean eccentricity $\overline{e}$ of 0.744. The fractal dimension $\delta$, a metric of shape complexity relating cloud perimeter to cloud area (Lovejoy, 1982) has a relative low value $\delta = 1.19$, indicating that

trade wind cumuli shapes are rather smooth and compact (as also found in Mieslinger et al., 2019). Accordingly, we can answer our question of a realistic cloud shape simplification by the elliptic fit. Despite persisting uncertainties in CTH, mainly in the range of 50 to 300 m, cloud area statistics are robust. After adding and subtracting the maximum CTH-uncertainties (Sec. 3.2), the slopes of the distribution remain comparable.

Overall, the viewing geometry from the 2D imager has significant impact on horizontal cloud size statistics. Circular assump-

tions from the along-track, instead, lead to stronger deviations in the distribution. The maximum along-track cloud length overestimates cloud sizes and the decay of the distribution becomes weaker. Using radar resolution, statistical methods, e.g. considering circular assumptions (Romps and Vogelmann, 2017) or as Barron et al. (2020), fail in reproducing the actual dou-



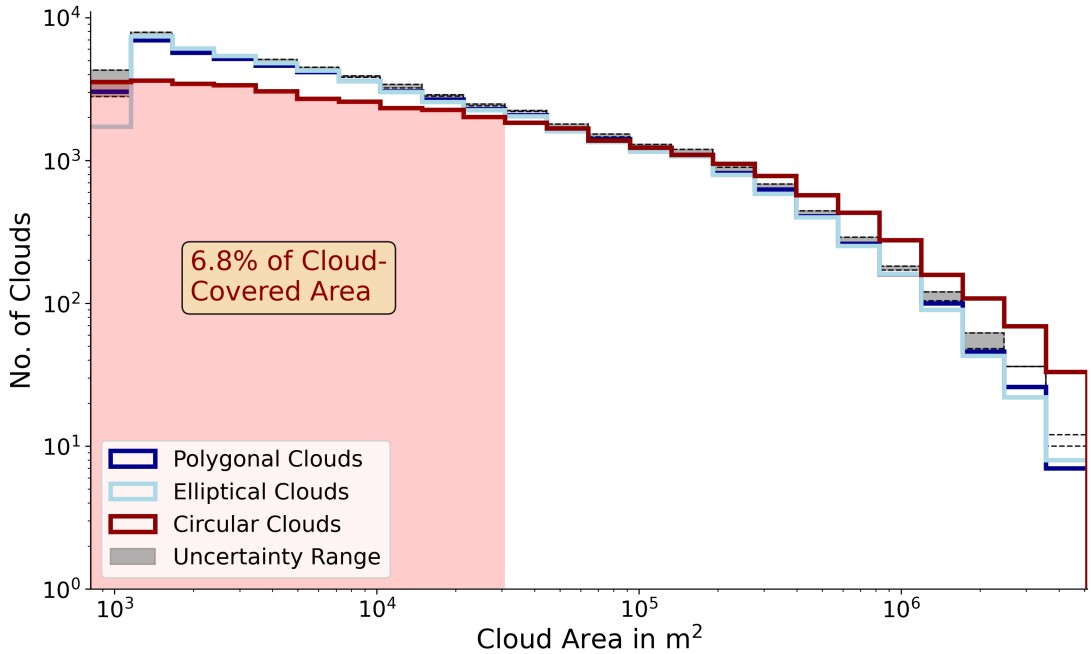

**Figure 10.** Shallow cloud area size distribution with absolute cloud number (No.) on the y-axis. The distributions of polygonal (dark-blue), elliptic (light-blue) and circular (dark-red) clouds based on the along-track length $L$, are compared. The red domain represents the radar sub-grid scale. The grey shaded pattern illustrates the variability due to the CTH uncertainty $U_{CTH}$ by adding $\pm U_{CTH}$ to each CTH.

ble power-laws (not shown). Cloud shapes being rather elliptical than circular underline the value of 2D observations by the imager. For clouds having a high eccentricity, cloud size statistics inferred from the along-track length can deviate substantially. The stronger the clouds undergo a directional elongation, the higher the chance that along-track lengths do not consider their dominant lengths and misinterpret the horizontal extension. On the other hand, the identification of factors regulating cloud orientation can help to optimize flight segments in a way that they align with dominant cloud elongation.

## 5 Cloud elongation under the impact of trade winds

This section refers to the elongation of clouds, where the wind field and its impact on 1D horizontal cloud geometries receives attention. In particular, trade-wind cumuli driven by surface evaporation do not rise fully vertically under dominant winds with vertical wind shear. As reproduced in LESs (Neggers et al., 2003; Helfer et al., 2020), this changes their horizontal projected area. Since wind speed is positively correlated to cloud deepening (Nuijens and Stevens, 2012) and to wind shear within the trade wind BL (Brueck et al., 2015), clouds can become more tilted with intensifying winds. By focusing on cloud's elongation for our application, we investigate to what extent the trade winds play a role in measuring horizontal cloud geometries using transect measurements. Considering RF03 and RF06, fifty dropsonde releases per flight provide a high-resolution spatio-





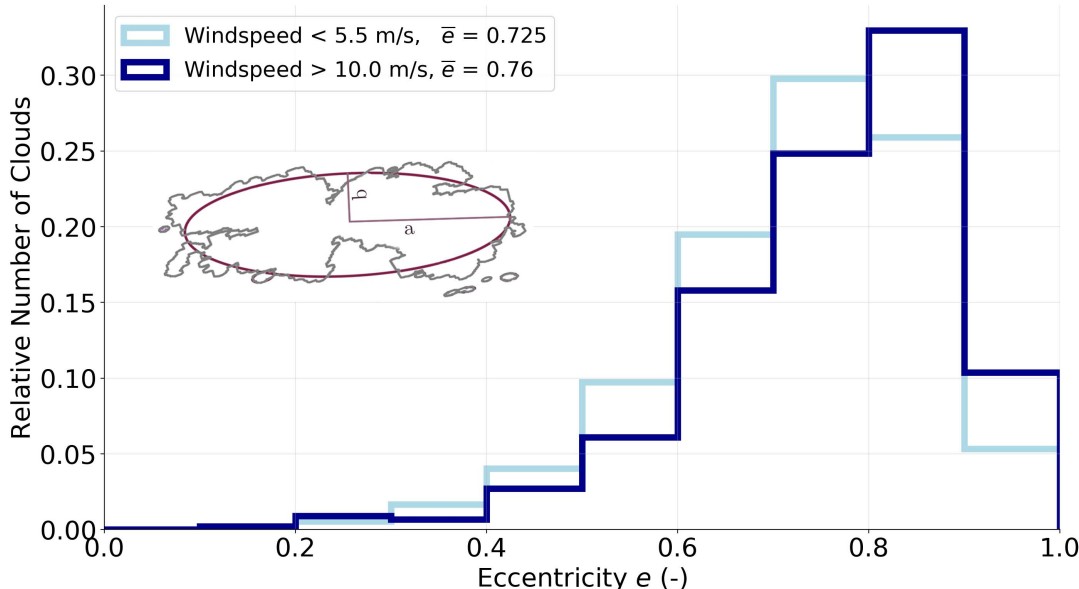

**Figure 11.** Eccentricity distribution of shallow clouds from RF03 and RF06 showing the relative frequency of clouds in eccentricity bins, having a width of 0.1. The samples are distinguished between low (below 5.5 m/s) and high wind speeds (above 10 m/s), and an exemplary elliptic fit showing major (minor) semi-axis $a$ ($b$) of the cloud.

temporal representation of the vertical wind field structure (Fig. 1). From that, we interpolate values between the profiles in time. Focussing on shallow cumulus clouds (CTH < 3000 m), the dropsondes reveal dominant easterly winds with wind speeds up to $12\,\mathrm{ms}^{-1}$ at CTH and vertical wind shear, between radar-based cloud base height (CBH) and CTH, up to of $3\,\mathrm{ms}^{-1}$.

We depict the eccentricities $e$ from the elliptic clouds (Sec. 4.2) and their frequency distributions for low and high wind
speed cases at CTH in order to elaborate relations between cloud elongation and increasing wind speed (Fig. 11). Low (high) eccentricities below (above) 0.5 indicate rather circular (elliptic) clouds. By considering eccentricities for wind speed thresholds aligned to 10th and 90th percentiles, both wind cases remain clearly distinguishable in their intensity and respective cloud sample sizes remain of similar size. Clouds shorter than 200 m are excluded, as their elliptic fits stem from poorly resolved cloud structures. Such very small clouds would show higher eccentricities due to irregularly sized imager pixels (Sec. 2.3).

We found both elliptic axes to differ by more than 10 % in length for 98 % of the clouds. Therefore, Fig. 11 shows eccentricities below 0.4 for less than 2 % of the trade-wind cumuli, regardless the strength of winds. Towards higher eccentricities, both histograms exhibit different characteristics. Stretching of clouds intensifies with increasing wind speeds. The maximum relative number of clouds is located around $e = 0.75$ for low wind speeds and around $e = 0.85$ for high wind speeds. At weaker winds, more clouds have eccentricities below 0.8. The mean eccentricity increases from $\overline{e} = 0.725$ at low wind speeds to $\overline{e} = 0.76$ at
higher wind speeds. Around 45 % of the clouds have eccentricities higher than 0.8 in the strong wind case.

If all along-track cloud observations relied on the semi-minor axis of elliptic clouds, mean maximum underestimations of dominant cloud lengths would be 31 % at low and 35 % at high wind speeds according to the mean eccentricity. This arises





the question whether or not the elliptic clouds undergo predominant orientations. Since cloud stretching seems to increase under stronger winds, it is hypothesised that clouds, i.e. their observed cloud tops, tend to tilt into the prevailing wind field.

If so, flight headings perpendicular to wind direction may cause underestimations in determined cloud lengths converging the worst-case scenario, being larger than 30 %.

In both wind scenarios, the cloud lengths in wind direction yield a mean underestimation of the actual major axis length by 21.7 % for low wind speeds and by 17,2 % for high wind speeds, respectively (not shown). For 40% of the clouds under high wind speeds and for 30% at low wind speeds, the dominant axes are underestimated by less than 10%, so that wind-induced

cloud orientation presumably comes into play. However, these values are ambiguous to quantify the effective cloud orientation into wind direction. Therefore, we define an eccentricity adapted normalised radius $r^*$ for each cloud:

$$r^* = e \cdot \underbrace{\frac{R_{\text{WindDir}}}{a}}_{r},\tag{3}$$

where $e$ is the eccentricity, $R_{\text{WindDir}}$ the length of the cloud ellipse in wind direction and $a$ the length of the semi-major cloud axis (as sketched in Fig. 11). The eccentricity-adapted normalised radius $r^*$ combines the magnitude of cloud stretching and

of major axis tilting into wind direction and thereby includes counteracting effects; If the axis length in wind direction for one cloud is similar to its major axis length (e.g. normalised radius of $r = 0.9$), but its eccentricity small (e.g. $e = 0.1$), $r^* = 0.09$ will reveal a weak effective orientation in wind direction. If a stretched cloud ($e > 0.9$) is also strongly tilted into the wind direction ($r > 0.9$), $r^*$ will exceed 0.81. The maximum effective orientation is given by $r^* = 1$.

Considering the frequency distribution of $r^*$ for both wind cases (Fig. 12), the product of $e$ and normalised radius $r$ yields

skewed distributions with a local maximum near $r^* = 0.55$ for both wind scenarios. When clouds are exposed to stronger winds, a higher proportion of clouds having an elongation into the wind direction exists. At stronger winds, the distribution shifts rightwards so that the mean $\overline{r^*}$ increases from 0.55 to 0.62. Looking at stronger orientated clouds ($r^* > 0.6$), the relative number of clouds absolutely increases by more than 10 % under high wind speeds. This means that are more stretched and their major axes tilt more intensely into wind direction under increasing winds clouds. Consequently, it becomes more likely

that potential sources in erroneously estimating major cloud lengths from along-track observations increase at stronger winds and increasing wind shear, as clouds elongate into the wind direction. Even though these results depend on the downward projections in the coordinate transformation (Sec. 3.2), they remain robust after including the CTH uncertainties (not shown). Besides deepening of non-precipitating clouds under stronger winds, as found in LES (Nuijens and Stevens, 2012) and observations (Mieslinger et al., 2019), various cloud-wind interactions are widely investigated in literature. Simulations in Bretherton

et al. (2013) show clouds reaching a certain depth until precipitation becomes more efficient with increasing surface-near wind speed. Recent LESs (Helfer et al., 2020) reveal signs of increasing wind shear acting as limiting factor on cloud depth with increasing wind speed. In our cases, the radar shows an increase of mean cloud depth of 30% at higher wind speeds. This is, however, more associated with CBH lowering instead of rising CTH. During NARVAL-II, the imager shows less evidence of increasing horizontal cloud size or of scale break shifts with increasing surface wind speed, as stated in Mieslinger et al. (2019).

Our variability in wind and geometry responses are, in turn, notably smaller and cases from additional campaigns might be beneficial. This may also further classify our tendencies of wind-induced cloud elongation.





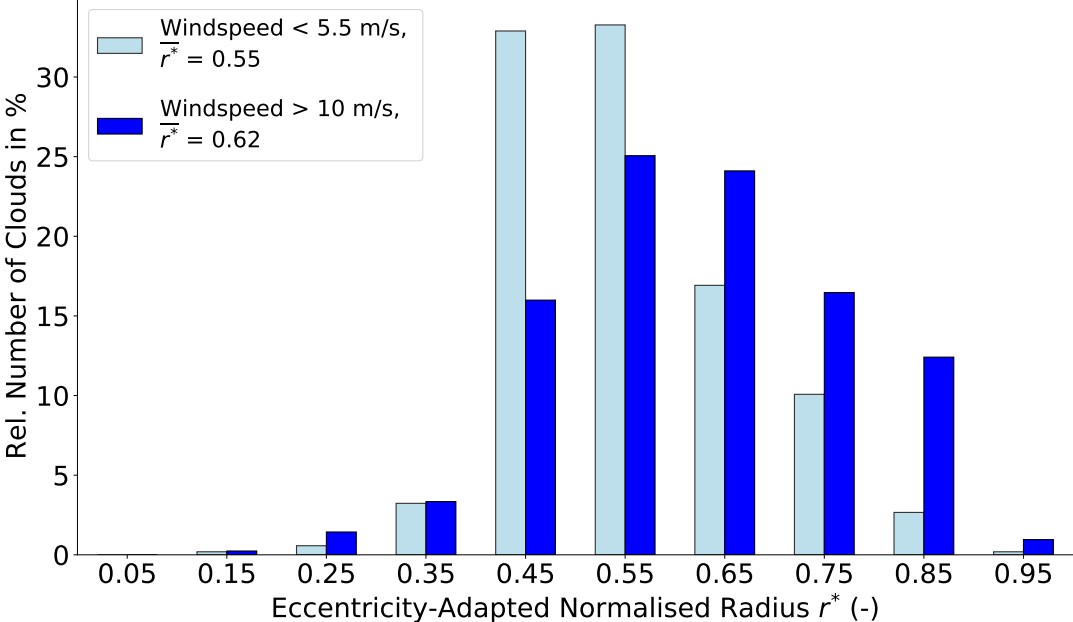

**Figure 12.** Effective orientation of shallow low-level clouds (CTH < 3000 m) from RF03 and RF06 described by the eccentricity adapted normalised radius $r^*$ for low wind speeds below 5.5 m/s (light blue bars) and high wind speeds above 10 m/s (dark blue bars) in CTH. Further specifications are given in the legend.

## 6 Conclusion and outlook

While the lower end of trade-wind cumuli size spectra is becoming resolvable, current research aims to link horizontal cloud geometries to cloud controlling factors in the vicinity of the marine BL (Mieslinger et al., 2019). However, observations from

vertical transects, such as by profiling radar that overpass clouds (space-/airborne) or that let clouds pass over (ground-based), have to derive 2D horizontal cloud geometries from their 1D along-track transect. Using state-of-the-art airborne devices aboard HALO, we compare statistics of horizontal cloud geometries from along-track radar transects with synchronous observations from the 2D imager specMACS. Combining their cloud masks, we stepwise identify impacts on cloud size statistics that arise from refining hectometer resolution (common scale for airborne profiler) to decameter (imager) and determine impacts

occurring from 1D to 2D nadir perspectives. Simplified cloud shapes, which maintain the cloud size statistics, facilitate the quantification of cloud aspect ratios and can reveal limits of approximating 2D geometries from transects measurements. Our major research questions set to be answered are:

– What remains unresolved in statistics of horizontal trade-wind cumuli geometries when using radar profiler having hectometer along-track resolution?

– How representative are cloud sizes derived from 1D transects after considering a nadir 2D FOV?





- Which simplified shape approximates horizontal cloud projections?

- To what extent do strong trade winds and corresponding cloud elongation play a role in assessing horizontal cloud sizes using transect measurements?

Although the cloud masks of the imager and radar provide along-track cloud lengths initially, horizontal 2D cloud geometries require their synergy. This includes a transformation of the imager cloud mask into a local coordinate frame with distance based axes, that requires the radar-detected CTH. Furthermore, using simplified cloud shapes, we investigate the coupling between clouds and BL trade-wind conditions in view of impacts on geometries inferred from both perspectives (1D and 2D) in predominant shallow convection. Frequent releases of dropsondes thereby allow us to characterise prevailing trade winds. This framework answers our before addressed questions as follows:

- Based on the imager cloud mask, the majority of clouds (more than 60 %) is subgrid-scale (along-track size < 300 m) for the radar. By coarsening the imager resolution to that of the radar (30 times coarser), we found around 80 % of single cloud objects to disappear or to merge. Such losses in sample size have an impact on along-track cloud size distributions. While, according to the imager, the cloud length distribution underlies double power law characteristics with a pronounced scale break around 500 m, this is not captured when remaining in radar resolution (1 Hz). However, it is worth-mentioning that such subgrid-scale clouds contribute to less than 10 % of cloud covered area.

- Profilers require extended paths to reconstruct along-track cloud statistics from imagers. The 2D imager FOV increases the amount of observed clouds by a factor of around 12. Furthermore, transect observations coincidently sample cloud edges. To have a proxy for arising errors, we consider the maximum along-track length within the 2D FOV and found that this varies slopes of along-track length distributions, albeit less the scale-break. The imager-based cloud area size distribution can limitedly be mirrored when approximating horizontal cloud extension circularly from along-track lengths.

- The 2D FOV reveals that observed trade-wind cumuli typically have a roughly elliptic shape. Since trade-wind cumuli are mostly compact in shape and consist of a certain aspect ratio (roughly 3:2), an elliptic fit constitutes a valid simplification in most cases that maintains the cloud area distribution (a double power-law). Elliptical clouds are not always orientated along the flight heading, so that along-track lengths tend to misinterpret the effective cloud length.

- Eccentricities for compact clouds show wind induced directional cloud elongation. At higher wind speeds, correlated with increased wind shear, clouds are slightly more stretched and tilt more effectively into the wind direction at CTH. On average, the cloud length in wind direction underestimates the actual major axis length by 17 % in the vicinity of strong winds ($> 10\,\mathrm{ms^{-1}}$) against 22 % at weaker winds ($> 5.5\,\mathrm{ms^{-1}}$). One-dimensional trade wind cumuli size statistics are thus not necessarily invariant to direction and are supposed to be affected by the wind field. Along-track size spectra can be biased as they do not always include the dominant cloud axes. For stronger winds, we expect circular assumptions from curtains to overestimate cloud areas by up to 50 %, if clouds are detected at their centre.

Our derived airborne cloud geometries are very similar to those stated from satellite observations (Mieslinger et al., 2019), highlighting the representativeness of the trade-wind cumuli captured during NARVAL-II. Remarkably, the imager captures 80



times more clouds than the radar. While only 2D observations enable the horizontal cloud extent to be determined without using

statistical assumptions, even along-track cloud sizes are very sensitive to the refinement from hecto- to decameter resolution and the observation FOV. Nonetheless, the abilities of profilers to separate overlaying clouds can prevent erroneous cloud geometries in imager observations. Thus, redundancy in cloud observations is fundamental.

Our quantitative assessment may serve as a benchmark that can be similarly transferred to other air-/ spaceborne observations and improve our knowledge of statistical representativeness from transect profiling. The results complement recent LES studies

transferring from 1D chord lengths to actual 2D cloud sizes (Barron et al., 2020). The dependency of cloud geometries on wind speed found in this study is weaker than expected, but robust. Results may become more unambiguous if more periods of collocated measurements and higher wind variability can be considered. Pursuing the idea of elliptic cloud orientation, we postulate uncertainties in along-track cloud sizes to increase more under stronger winds. In turn, future transect studies should consider that extended flight paths along strong winds provide cloud sizes that are most representative for their dominant

lengths. Limitations in our analysis of 2D cloud geometries occur where the coordinate transformation (Sec. 3.2) deals with high uncertainties of across-track CTH. For that, stereographic methods, e.g. Kölling et al. (2019), and synergies distributing nadir information from radar to the across-track pixels, specified in Barker et al. (2011) and optimised in Höppler et al. (2020), provide additional CTH estimates extending suitable flight periods. This raises new capabilities to examine impacts of cloud internal wind shear on three-dimensional (3D) cloud processes, as seen in LESs (Helfer et al., 2020).

Furthermore, the HALO campaign *Elucidating the role of clouds-circulation coupling in climate (EUREC[4]A)* specified in Bony et al. (2017) and Stevens et al. (2021), which was conducted early 2020 and equipped with the same devices aboard HALO (Konow et al., 2021), adds a large set of tropical marine cloud observations. EUREC[4]A data, which are currently being processed, comprise measurements during the dry season and allow a seasonal comparison. Due to increasing subsidence rates and lower tropospheric stability, shallow convection is therein more favoured than during NARVAL-II within the wet

season (Medeiros and Nuijens, 2016; Stevens et al., 2016). Stronger trade winds generally increase low-level cloudiness further (Brueck et al., 2015). By broadening the range of BL conditions, EUREC[4]A will considerably increase the framework to analyze coupling between cloud geometries and trade winds. Applying our elliptic approaches, it remains to be seen whether the wind affects cloud geometries more strongly during EUREC[4]A or whether different wind-induced mechanisms on shallow cloud geometries occur. Certainly, the potential in analysing the coupling between trade-wind cumulus cloud geometries and

the BL is far from being exhausted. A better understanding of trade-wind cloud geometries and their link to cloud-controlling factors using airborne remote sensing, such as profiler and imager, may help to lower uncertainties of shallow cumulus cloud feedbacks in current climate models.

*Data availability.* Both cloud datasets from the NARVAL-II campaign are freely accessible. The specMACS cloud mask data are available at the macssserver https://macssserver.physik.uni-muenchen.de/campaigns/NARVAL-II/products/cloudmask/ (accessed at: 05 October 2021).

The processed radar measurements used in this study are published and freely accessible under Konow et al. (2018).





*Author contributions.* FA and HK were main initiators for the work in the scope of this manuscript. Using specMACS data provided by the working group "Remote Sensing and Radiative Transfer" at the Ludwig-Maximilians-Universität München (LMU) under the lead of B. Mayer, HD conducted the analysis presented and created the text under scientific support of FA and HK.

*Competing interests.* The authors declare that they have no conflict of interest.

*Acknowledgements.* This study was partly supported by the Deutsche Forschungsgemeinschaft (DFG; German Research Foundation) under the HALO SPP 1294. Moreover, we want to explicitly thank T. Mieslinger, T. Kölling and L. Höppler for various helpful discussions. Special honour is addressed to F. Gödde and the entire team of the working group "Remote Sensing and Radiative Transfer" at the Meteorological Institute of the LMU for providing the specMACS data. Thanks also go towards Norbert Noreiks for delivering sketches of the research aircraft.





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
