# Peer review of "Horizontal geometry of trade-wind cumuli – aircraft observations from shortwave infrared imager versus radar profiler"

_Atmospheric Measurement Techniques, 2021_

## Author Response (AR2)

**Point by Point Response for Manuscript, entitled ́Dorff, H. et al. 2021: Horizontal geometry of trade-wind cumuli – aircraft observations from shortwave infrared imager versus radar profiler', for peer-review submission**

**Table of Contents**

**I) Response and Changes to the comments from Anonymous Referee 1**

We thank the AMT associating editor, Maximilian Maahn, as well as the Anonymous Referee #1, for this enlightening review. Please find below our responses (in standard font) to the remarks from the Anonymous Referee #1 (in *italics*), which were given analogously in the letter of responses uploaded before. The changes/ modifications in the manuscript are specified below. Lines given refer to the lines in the revised manuscript.

1. *Is AMT the best venue for this paper? Sure, it is nominally about comparing different observational techniques, but the results are much more broadly applicable to the cloud physics community. I feel ACP would be the more appropriate journal in the EGU stable. I therefore would suggest **moving to a different journal,** but otherwise **minor revisions***

**Response:** Indeed, our manuscript deals with a variety of characteristics and metrics relevant for better understanding of cloud physics. Since we link our findings to boundary layer conditions (primarily the wind field), we can understand your suggestion for another journal specialized on cloud physics and strongly thank you for this advice. However, we intend to approach the overall topic of trade-wind cumuli geometries from the observation perspective and in particular, from how the measurement methods can deteriorate our understanding of prevailing cloud geometries. We propose future work considering our technical methods to investigate the interactions of cloud geometries under the impact of the trade-wind boundary layer in more detail. A follow-up study using EUREC4A data, which extends the cloud sample significantly together with a well-defined characterisation of the boundary layer by dropsondes might be a helpful contribution to, e.g., ACP.

2. *In Section 4.1, I would like to see a few more statistics about the compatibility between radar and imager. For instance, what is the % agreement between the two on clear/cloudy pixels (false positive/negative rate, if you will). Is this a function of certain parameters and choices of thresholds?*

**Response:** We determined a rated matrix according to your suggestion including true positive rate (TPR), false negative rate (FNR), false positive rate (FPR), true negative rate (TNR), where we refer radar to regridded imager. We use this convention for clarity without assuming that the imager reflects necessarily the truth. We compile one matrix for the entire period (Table 1) and one (Table 2) for the flights RF03 and RF06 only, which were dominated by shallow convection.

*Table 1: Rate matrix for clouds seen in radar and imager for all collocated flight periods.*

| % | radar cloud | no radar cloud |
|---|---|---|
| imager cloud | 77.94 (TPR) | 22.06 (FNR) |
| no imager cloud | 7.41 (FPR) | 92.59 (TNR) |

*Table 2: Rate matrix for cloud pixels seen in radar and imager for shallow convection dominated flights (RF03, RF06).*

| % | radar cloud | no radar cloud |
|---|---|---|
| imager cloud | 32.83 (TPR) | 67.16 (FNR) |
| No imager cloud | 3.30 (FPR) | 96.70 (TNR) |

In fact, the radar is clearly less sensitive and detects only 78% (33%) of the all (shallow convective) clouds. At the same time, there are hardly any clouds, which are observed by the radar only. Just the morphological closing applied to the radar cloud mask may create faulty clouds. This effect results in a small false-positive rate of 7%(3%) for all (just shallow convective) clouds.

Apart from predominant cloud types, the results depend on the thresholds for assigning coarse gridded imager pixels as cloudy or clear (set to 50% in Sec. 3.3). Figure 1 illustrates how the device differing results (FPR and FNR) depend on this threshold for the specific flights.

[Figure]

*Figure 1:a) Ratio of no radar cloud pixels to cloud pixels in regridded imager (false negative rate) and b) ratio of radar cloud pixels to no regridded imager cloud pixels (false positive rate) for given threshold of imager pixels being cloud in high resolution for one single pixel in coarse-gridded resolution.*

Increasing the threshold for the regridded cloud masking, the false negative rate decreases for all flights (Fig. 1a) showing that lower sensitivity in coarse-gridding creates more cloud gaps as seen by the radar. However, in particular for RF03 and RF06, the false negative rate remains significantly above 50 % as many clouds are not only fragmented but also completely undetected by the radar. Between the flights, the false negative rates show large spreads, while the false positive rate varies much less between flights and thresholds (Fig. 1b). For flights in shallow convection (RF03 and RF06), false positive cloud pixels from radar exist for less than 5% for all masking thresholds.

In conclusion, we see a higher dependency of radar based false negative rate on cloud types and on regridding thresholds as for the false positive rate. At this point, we want to remind the reader that we are using an imager cloud mask, which is also dependent on various thresholds. Altogether, this supplement analysis underline the complexity in how to determine the amount of clouds at a given region even when using collocating measurements. Since the here presented plots focus more on impacts on cloud fraction without considering individual cloud objects, this supplementary study is

slightly above the scope of this manuscript where we want to focus on the representation of coherent cloud objects and their geometries. Nonetheless, we really appreciate your suggestion so that we included the following sentences in our manuscript at the beginning of Section 4 before coming to the coherent cloud objects:

**Modifications in Manuscript (Line 239):** "Comparing cloud fraction from regridded imager and radar after the FOV adjustment (Sec. 3.1), the radar is clearly less sensitive and detects only 78% (33%) of the all (shallow convective) clouds. At the same time, there are hardly any clouds, which are observed by the radar only. Just the morphological closing applied to the radar cloud mask may create faulty cloud pixels. This results in a small false-positive rate of 7% (3%) for all (just shallow convective) clouds. The rates also depend on our set regridding threshold (Sec. 3.3) but to a lesser extent. Evidently, the mentioned differences in cloud fraction affect single cloud geometries."

*b) Do the 1D CSDs from both instruments pass a KS test for certain sets of parameters?*

**Response:** Indeed, the KS-test represents a very useful statistical tool to compare the two samples (regridded imager and radar cloud size) and their distribution. We used different viewing curtains of the imager (light lines in Fig. 6) to indicate how the distribution varies for different viewing angles. Their envelope shows the robust difference between the radar distribution compared to the coarse-gridded imager for the entirety of imager viewing angles. This difference originates especially from the unresolved shallow low-level clouds leading to a higher relative contribution of larger clouds in the radar-based distribution. This effect partially decreases by increasing the cloud mask threshold when regridding. However, we cannot detect cases where the KS test will reveal an equal probability distribution.

3. *I noticed some choices of words where I am not sure I would have given those words the same meaning. Some suggestions for alternatives are below whenever I found them – not exhaustive, and maybe not always what you intended to say. It would be good to go through the paper with a non-native reader in mind with a somewhat limited English vocabulary, and when in doubt just use the simplest words possible. Otherwise the paper is written in a very clear language.*

Thank you for your advice regarding wording and phrasing. In the following, we specify our adjustments in the revised manuscript according to your suggestions.

*Minor comments and word suggestions:*

*L 14: While-> Since*

**Response/Modification (Line 14):** the word has been deleted but its sentence was also rephrased (see next response).

*L15: Do clouds become invisible, or simply gridpoint? The lower end of your CSDs is not much discussed, other than by the scale break. I can see several different mechanisms at play here*

**Response:** We modified the sentence.

**Modifications in Manuscript (Line 14):** The radar encounters difficulties to represent clouds shorter than 200 m as they are either completely unresolved or considered like single grid points. Very shallow clouds can also remain unresolved due to a too low radar sensitivity. Both facts deteriorate the cloud size distribution significantly at this scale.

*L42: Why is that a challenge? If anything, perhaps "2D imagers are better equipped to address the challenge…" or so*

**Response/Modification:** We changed the sentence accordingly in line 44.

*L50 barely -> rarely.*

**Response/Modification:** we replaced the word in line 53.

*L56 LES has been able to do this for a while, but now also for large domains (>100km)*

**Response/Modification:** We added this for clarification at this place in line 59.

*L73: How collocated are the instruments? How many meters away in spanwise and streamwise direction? I doubt that at least the streamwise direction is going to matter much (after the correction you talk about later), but good to mention here either way.*

**Response:** Yes, this is a crucial aspect. The devices have fixed viewing directions on the aircraft frame and hence also between each other. As shown in Fig. 2,4,7,9, the orange line, indicating the radar FOV, is not centered in the imager FOV, but slightly shifted. In specific, the imager looks 2.6° ahead of the radar. The central imager across-track pixel is located by 0.55° shifted to the left in flight direction. For the modification of the manuscript, we have the impression that giving all these details in the introduction can be a bit distracting for the reader. Therefore, before going into detail in Section 3, we briefly referred to it in line 71 as follows:

**Modifications in Manuscript:**

**Line 71:** "Section 3 encompasses our applied synergy of both cloud masks comprising FOV adaptation between both devices and coordinate transformation in order to […]."

Moreover, we introduced Sec. 3.1 (Line 147) by: "When comparing both devices in their overlapping transect, one has to consider that the central imager across-track pixel is 0.55° shifted to the left in flight direction referred to the radar. Moreover, the imager FOV is 2.6° ahead of the radar, so that both devices look at identical clouds at different time frames."

*L99: How much is this FOV in practice in meters? and what is the typical resolution in meters? I'm not a fan of pixel# as a unit. Perhaps the spatial equivalents of Time and Pixel units can be put on secondary x/y axes in figs2,4,7,9?*

**Response:** Typical resolutions are given in Sec 2.3, but still we fully understand your impression and suggestion regarding the plot axes. However, adding secondary distance-based axes leads to issues for the shown illustration of the imager because the aircraft has a changing air attitude (pitch, yaw, roll angle). Moreover, the distance

between signal source (cloud or ocean) and aircraft changes at every point. Requiring the coordinate transformation for each cloud (Sec 3.3) as done in later analysis, the squared panel in the mentioned figures become snappy at the borders when showing the transferred cloud mask. At this place, we consider this as inappropriate for the readability.

**Modifications in Manuscript Table 1:** We now clearly mention the FOV pixel values in Sec 2.2 for the entire imager FOV (line 104), and the single pixel FOV in Fig 2, 4.

*L102: pronounce -> result*

**Response/Modifications:** wording changed in line 107.

*L105 non-zero reflectivity*

**Response/Modifications:** declaration adapted (line 110).

*L143: Is CTH the correct metric? Since you're integrating over the entire depth of the cloud, mid-cloud level would be more precise, I guess. Again, shouldn't matter much in practice for these shallow clouds.*

**Response:** We confirm that our method possesses weaknesses. They result from the restricted abilities we have. Principally, the imager sees clouds from the top, and the horizontally projected cloud top reveals the cloud shapes in the 2D mask. Accordingly, CTH is the correct metric, although at the edges of the FOV clouds are slightly more captured from the side as described in Sec. 3.2.

However, indeed, a certain cloud penetration path exists for the imager and radar. Comparing with simultaneous LIDAR measurements (e.g. Gutleben et al, 2019), we have found tendencies of the radar to underestimate cloud top height slightly. In particular, for the shallow low-level clouds, radar-based CTHs (as a function of time) are supposed to be too low due to the radar sensitivity, so that the signal actually originates from slightly deeper inside the cloud a little bit towards mid-level height.

Therefore, we conducted various visual inspections of the collocated cloud masks such as we illustrate in Fig. 4. They show us an overall reasonable time shift being adapted to the radar for several cloud cases. Furthermore, when considering the uncertainties of CTH (Sec. 3.2), our results remain robust.

*L143: "lower and further to the aircraft" not sure what that means exactly.*

**Response/Modification:** We changed this in line 149: "The lower and more distant the clouds are to the aircraft, […]"

*L149: Emphasize 2D connectivity*

**Response/Modification:** the emphasis on two dimensions is now given in line 154.

*L174: dammed -> limited*

**Response/Modification:** We replaced the word in line 181.

*L183: This does introduce the bias that cloud size is artificially limited by the FOV size.*

**Response:** Yes, it truly does. This explains why we also compare all along-track cloud sizes at least to lengths up to 10 km when remaining in the aircraft-following coordinate system without neglecting incomplete clouds in Sec 4.2 and 4.3. Both approaches have their pros and cons. To make your point clearer, we rephrased the bullet point as follows:

**Modifications in Manuscript (Line 190):** "Clouds extending out of the FOV are neglected as their 2D geometries exterior of the FOV are not obtainable. In addition, we exclude clouds having an along-track cloud size bigger than the imager across-track FOV, as otherwise large clouds orientated along the flight path would be considered preferentially. Since cloud size then becomes artificially limited by the FOV size, cloud size statistics remain biased for undersampled larger clouds approaching the typical image scale.

*L240: This may be cloud misrepresentation, but it is the fair comparison between the two instruments. This is an important part, because it validates the radar for use in the (extremely common) situation that no imager is available.*

**Response**: We agree that this is a fair comparison, which is why we considered the overlapping 1D path in the following sections from both devices. To make our statement on the effects from 2D to 1D more understandable, we rephrased the second last sentence of the paragraph as follows:

**Modifications in Manuscript (line 255):** "[…] fraction of ID labelled clouds from the 2D field reaches into radar curtain (Fig. 5). Otherwise, cloud fragments can appear distinctively within the 1D curtain although belonging to one cloud. Neglecting the across-track, such fragments lead to a considerably different sample (higher cloud amount) and cloud size misrepresentation.

*L281: I would be interested in a bit more discussion of the scale break, as it is located much sooner than often reported for shallow Cu (1km+). Are the authors sure that this is not an artifact of the observational strategies/instrument resolution?*

**Response:** Thank you for mentioning the complexity in the interpretation of the scale-break. Although there exist studies showing higher scale-break values, we found several studies considering observations of comparable resolution that suggest plausibility of our results. In our literature review we conducted before, we found various sources, that resolve clouds below hectometer scale, locating scale breaks between 0.5-1.0 km (such as Zhao and Di Girolamo, 2007; Dawe and Austin, 2012; Heus and Seifert, 2013). Another source we want to highlight in this regard is Mieslinger et al. (2019). Using spaceborne observations from ASTER having comparable resolution to the airborne imager, Mieslinger et al, (2019) so far comprise the largest dataset of marine shallow cumulus clouds covering also our region of interest. Since the dataset of Mieslinger et al (2019) includes decameter resolution for domain sizes in the order of hundred kilometers, we put a lot of trust in their findings.

Higher values of scale-breaks with 1km+ mostly occur for dataset constrained to coarser resolutions (e.g. Wood et al, 2011) not capturing the ubiquity of very small-scale clouds, which may then be merged to single hectometer scale cloud objects. Yet, we agree on your remark highlighting the complexity of scale break location and its origin that might be atmospherically driven or method-specific, why literature keeps on actively debating.

**Modifications:** In our manuscript, we gave some respective information in Sec. 3.4 beforehand and afterwards in Sec. 4.3 when considering the entire 2D FOV with a larger cloud sample. We have the impression that it is more appropriate to add some final discussion of the scale-break at the end of 4.3. Therefore, we added to line 298: "We discuss its location further in the following section when considering the entire 2D FOV with increasing cloud sample size."

In Sec. 4.3, we included more literature to compare scale break values. In accordance with the remarks of *RC2*, we added the following after line 346:

"Thus, observations from the imager specMACS during NARVAL-II well reproduce trade-wind cloud size distributions found in comparable studies.

Nonetheless, we highlight the ongoing debate in literature about location and artificial or boundary-layer driven origin of scale breaks (Mieslinger et al., 2019). We see that resolution affects the location of the scale-break in a way that it is missing for airborne observations in hectometer resolution. Although Wood and Field (2011) locate scale breaks above 1 km using hectometer scale spaceborne data, we cannot identify this from the radar curtain samples. On the other hand, larger clouds may be misinterpreted from the imager if only their edges reach into the imager FOV and artificially enhance the scale break through length underestimation. If we completely neglect clouds reaching out of the FOV, we do also produce biases with increasing cloud size (Sec. 3.2). Due to the complexity of scale break origin, some studies, e.g. van Laar et al. (2019), suggest to apply exponential power law fits (Ding et al, 2014) to prevent the scale break by a modified single distribution. "

*L298: Could be interest to compare the overlap corrections from Sulak et al (JGR, 2020).*

**Response:** We agree that investigating the cloud overlap ratio from the radar is applicable and of certain interest. Unfortunately, we are not certain where to place this source at L298 as we completely remain in the horizontal projection and do not consider vertical overlap, here. Yet, as your feedback is inspiring, we included an average horizontally inverse length factor defined as the average ratio of maximum along-track length divided by effective length in the radar-equivalent imager curtain (red distance divided by blue distance in Fig.07). We include this in the manuscript as follows:

**Modifications in Manuscript (line 317):** For all imager clouds reaching into the radar curtain, we calculate a mean inverse length factor, defined as the average ratio of maximum along-track length divided by the effective cloud length in the radar curtain (blue and red distances in Fig. 7). For these clouds, this horizontal analogue to the vertical overlap ratio in Sulak et al. (2020) reveals that, on average, maximum along-track cloud length differs to curtain length by a factor of 2.63. Since this factor has strong dependency on cloud length, its value strongly varies with cloud size and is presumably underestimated due to the FOV limits for larger clouds."

We further included some suggestions for complementary studies following Sulak et al. 2020 in the outlook, as given: "With this 3D representation based on merged imager and profiler, inverse cloud overlap studies such as from Sulak et al. (2020) can be complemented for vertical and horizontal perspective under the influence of BL characteristics derived from dropsondes."

**II) Response to the comments from Anonymous Referee 2**

We thank the AMT associating editor, Maximilian Maahn, as well as the Anonymous Referee #2, for this enlightening review. Please find below our response to the comments (in *italics*) from the Anonymous Referee #2.

*I have a suggestion which is not necessary for the authors to perform, but I wonder why they did not show at the end a joint-frequency distribution of cloud size and eccentricity? Should we expect some trend in eccentricity with cloud size? Perhaps eliminate figure 9 whose purpose is unclear, and add such a joint-pdf plot? Or leave that for your next work. There are other possible follow-up studies, which just illustrates why this is a good piece of technical foundation.*

**Response:** We cordially thank you for your summarizing feedback of our manuscript and hope that also others enjoyed reading it giving new perspectives for further cloud geometry analysis.

**Response to Joint distribution:**
Figure 1 illustrates our version following your plot suggestion:

[Figure]

*Fig. 2: Joint-frequency distribution of cloud size (referred to area-equivalent diameter) and eccentricity for clouds of RF03 and RF06 with L>200 m and the cloud samples affected by low and high wind speeds. Contour lines in major panel represent three iso-proportions of the kernel density estimate (KDE), distributions for both quantities are given separately at top and right.*

First, we really appreciate your suggestion, which is a truly interesting insight on cloud geometry interactions with the wind-field. We inserted the before shown Fig. 1 as Fig. 12 in the manuscript between the discussion of eccentricity and cloud orientation by

the normalised radius. Our analysis revealed the following that is now included into the manuscript:

**Modifications (line 419):**

"We hypothesize that larger clouds tilt more into the winds and become even more stretched. For the given clouds of RF03 and RF06, Fig. 12 illustrates the joint-frequency distribution of eccentricity and cloud size also depicting both wind cases. Referring to area-equivalent cloud diameters, the cloud size distributions remain comparable between both wind cases (top panel of Fig. 12) in contrast to an increase of horizontal cloud size and shifts of the scale break with increasing surface wind speed stated in Mieslinger et al. (2019). Inspecting the contours of the joint-frequency distribution, we identify an increase of eccentricity with cloud size especially for higher wind speeds. However, since the sample size strongly decreases for larger clouds as they are rarer and more likely to extend out of the imager FOV, our hypothesis can only be proven by longer observation periods or broader FOV. We thus encourage corresponding studies using e.g. ASTER as in Mieslinger et al. (2019)."

**Minor revisions:**

Line 25 "limitedly understood" is strange phrasing, consider "only partially understood"
**Response:** we modified this accordingly. (line 27)

Line 50 "barely" should be "rarely"
**Response:** we changed the wording (line 53)

Line 72 "precedent" doesn't make sense here, maybe "The conclusions consider the abilities of a prospective flight campaign to answer new research questions."
**Response:** we amended the sentence (line 76)

Line 74 "We consider" change to "we analyze"?
**Response:** we modified this accordingly (line 79).

Line 78 "... in a 2D image."
**Response:** changed (line 83).

Line 85 "... the profiling radar enables"
**Response:** we adapted the grammar (line 90)

Line 96 Why choose the SWIR band instead of VNIR? Just one short sentence to explain. Something about sun-glint perhaps?
**Response:** We added the following information to the sentence:
**Modifications (line 103):** "[…] which allows more explicit cloud masking under the presence of sun glint"

Line 127 I wonder if the window freezing issue should be mentioned again at the end in the recommendations.
**Response:** The instrument provider has solved the issue by adapting the design. A recommendation is no longer necessary.

Line 164 "following aspects" should be "the following aspects"
**Response:** changed accordingly (line 171).

Line 174 consider "limited" instead of "dammed" which sounds too much like "damned"
**Response:** we changed the wording following your suggestion (line 181) .

Line 183-185 This is a very important and difficult aspect of this analysis, I have experience with this issue. I believe you made the best choice to reduce bias and error of counting a section of a large cloud as a small cloud. However, this does not eliminate bias, but shifts the bias to larger size clouds (which are now systematically undersampled), so that is ideal for this analysis focused on smaller scales, but maybe a note about this should be included in the discussion in the beginning of section 3.4 to warn future users of this analysis technique. The larger clouds are indeed more rare, but will also be undersampled as the cloud length scale approaches the typical image scale.
**Response:** Indeed both simplifications for larger clouds have pros and cons. This is why we show both simplifications for both coordinate frames, where the mentioned aspects in Sec. 3.2 apply for 2D cloud geometries after coordinate transformation and are not conducted within the along-track coordinate frame. We admit that we stated this distinction a little bit unclearly in the manuscript (which explains your remark for line 315). Therefore, we highlighted the differences at several lines in the manuscript:
**Modifications:**
First, to underline the relation of Sec 3.2 to only 2D cloud shapes and not along-track cloud sizes, we modified the section title from "Distance based pixels" to "Two-dimensional distance-based pixels" (line 160).
Second, we added the following note in Sec. 3.4 (line 228):
"As long as we remain in the aircraft following coordinate frame, considering along-track cloud lengths, no restriction in cloud cutting as suggested in Sec. 3.2 is performed to include a high amount of clouds. Nonetheless, we only show distributions for lengths up to 10 km, because we are limited by the imager FOV when considering 2D cloud shapes."
Third, we discuss at the end of section 4.3 (line 351): "[…] larger clouds may be misinterpreted from the imager if only their edges reach into the imager FOV and artificially enhance the scale break through underestimation. If we completely neglect clouds reaching out of the FOV, we do also produce biases with increasing cloud size (Sec. 3.2)."
In Sec. 4.4, we also remind the reader that only fully detected clouds are now included (line 359): "Hence, we put now emphasis on the misrepresentation of along-track lengths for the effective 2D cloud size of all shallow cumulus clouds fully detected by the imager."

Line 214 A "scale-break" might also be a sign that a power law is the wrong choice,because, for example, a scatter-plot of frequency vs length scale data on a log-log axis plot that looks like two power-laws with a scale break in between could instead be considered as a single exponential distribution, with the "scale-break" location being the bend in the exponential on a log-log plot. Since 1 function is less complex than 2 functions, the principle of parsimony would suggest considering an exponential distribution instead of a power-law. I don't expect you to change this for this paper, or change the power-law obsession everyone seems to have, but I do suggest that you consider the exponential instead of a "scale-break" in future work.

**Response:** We thank you very much for your enlightening comment regarding the complexity of scale-breaks. Indeed, this topic is widely discussed. As mentioned in the answer of RC1, we added some discussions in Sec. 4.3.

**Modifications (line 348)**: […] we highlight the ongoing debate of literature about location and artificial or boundary-layer driven origin of scale breaks (Mieslinger et al., 2019). We see that resolution affects the location of the scale breaks in a way that it is missing for airborne observations in hectometre resolution. Although Wood and Field (2011) locates scale breaks above 1 km using hectometre scale spaceborne data, we cannot identify this from the radar curtain samples. On the other hand, larger clouds may be misinterpreted from the imager if only their edges reach into the imager FOV and artificially enhance the scale break through length underestimation. If we completely neglect clouds reaching out of the FOV, we do also produce biases with increasing cloud size (Sec. 3.2). Due to the complexity of scale break origin, some studies, e.g. van Laar et al. (2019), suggest to apply exponential power law fits (Ding et al., 2014) to prevent the scale break by a modified single distribution."

Line 227 Unclear "this prounounces"?
**Response:** we changed this to "this affects" (line 237).

Line 276 "contrarily" is not clear, something more like "This affects the distributions in the opposite direction Caption of Figure 5 Remove "exemplary"
**Response:** We amended the wording as suggested (line 292).

Line 315 The point of Figure 9 is unclear... are you trying to show clouds that don't make it into the analysis at all? Maybe draw some lines on Figure 9 to indicate which clouds in that image are included (if any?)
**Response:** In Sec. 4.3, we rely on the aircraft following coordinate frame and along-track cloud sizes, which do not use the coordinate transformations defined in Sec. 3.2. We admit that this fact should be stated clearer before.

In Sec. 4.3, instead, we can calculate the along-track cloud size independently of any assumptions for e.g. cloud top height which would require profiler data. The length values are unambiguous, at least for the given imager sensitivity. This advantage motivated us to add this section before dealing with 2D cloud shapes in Sec. 4.4 where we then have to consider the bullet points of Sec. 3.2. Neglecting clouds reaching to the FOV edges is here not yet done although we then have the issue of underestimated cloud length (see comments above). Hence, Fig. 9 just intends to illustrate the aspects described in line 313-317 of the preprint.

**Modifications (line 337):** In order to achieve this more precisely, we included the radar curtain in Fig. 9. In addition, we rephrased the reference to Fig. 9 in a paragraph in order to make the purpose of the visual explanation clearer:

"Clouds exterior of the radar FOV increase the sample non-uniformly which can be seen for an exemplary cloud scene in Fig. 9. When clouds are larger in the along-track axis, they generally tend to be larger in the across-track as well, often reach to across-track edges covering the entire FOV and are embedded in a cluster of numerous small clouds around (Fig. 9). For measurements constrained on a narrow transect, this results in a lower chance of small clouds around to be captured, whereas the largest clouds (at 17:43:15 and 17:44:20 in Fig. 9) are detected anyway, regardless of the resolution. Transferring from 2D to 1D transects thus flattens the distribution."

Line 316 "constraint" should be "constrained" and "This pronounces" doesn't make sense, maybe "This results"
**Response:** we changed both following your suggestions (line 340).

Line 317 "regardless of the resolution"
**Response:** we corrected this part (line 341).

Line 352 Not clear, maybe change to "Using only the radar resolution and statistical methods, e.g., considering circular assumptions (Romps and Vogelmann 2017), or as in Barron et al. (2020), such methods will fail to reproduce the actual double power-laws (not shown). Cloud shapes being rather more elliptical than circular..."
**Response:** we changed the phrase according to your suggestion (line 387).

Line 382 "arises" should be "raises"
**Response:** we changed the wording accordingly (line 427).

**III) Additional author's changes**

This section lists changes that do not result from the comments of the reviewers. These changes in the manuscript are here highlighted **bold**. The given line numbers refer to the updated manuscript version. Here, we do no list all minor changes comprising slight changes in phrasing or slight shortenings, such as:

- "methods described in Höppler et al. (2020)" → "methods in Höppler et al. (2020)"
- "Hence, same clouds are hence", → "Same clouds are hence"

Such cases can at least be identified in the tracked changes file.

**Abstract:**

Line 19: We inserted the tendencies of cloud elongation enhancement with cloud sizes (joint-frequency plot, Fig.13) stating: Trade-wind cumuli show horizontal patterns similar to ellipses with a mean aspect ratio of 3:2 **having tendencies of stronger elongation with increasing cloud size.**

Line 20:"Instead of circular **cloud shape** estimations", to make this point clearer.

**Section 2:**

Line 86:  **Periods**

**Section 3:**

Line 161: we added the clause: **"and not only along-track specific"**

Line 162:  → **transformed**

**Section 4:**

The beginning of Section 4 was adapted:

- Line 236:  → **Varying sensor sensitivities and resolutions change cloud masks,**

Line 250: **"[…] radar and imager**  → **also influence horizontal cloud geometry statistic"**

Line 255: We cut this expression: **"Otherwise, cloud fragments can appear distinctively within the curtain although belonging to one cloud. Neglecting the across-track, such fragments lead to a considerably different sample (higher cloud amount) and cloud size misrepresentation."**

Line 297: "This scale break is located  →**slightly above radar"**

Line 311: Several changes in the phrasing: **With regard to the along-track coordinate frame, the 2D**  **perspective reveals another**

**representation of the along-track cloud size  by considering deviations in the across-track  direction, which is sketched for an exemplary cloud in Fig. 7.**

Line 323: We corrected this sentence grammatically and amplified its message:

**"Moreover, the across-track perspective includes additional cloud objects […]"**

Line 341: We added the transect. **"[…] any way, regardless of the resolution and 1D transect."**

Line 346: **we highlighted the similar resolution.**

Line 359:  To define the included cloud samples clearer, we included some information: **"Hence, we put now emphasis on the misrepresentation of along-track lengths for the effective 2D cloud size.  of all shallow cumulus clouds fully detected by the imager. The actual 1D cloud size projection"**

Line 386: **"The maximum along-track cloud length overestimates cloud sizes and the decay of the distribution becomes weaker  and is supposed to be even less when not neglecting clouds reaching out of the FOV (Sec.3.2). Using only the radar resolution and statistical methods, e.g. considering circular assumptions (Romps and Vogelmann, 2017) or  in Barron et al. (2020), such methods will fail to reproduce the actual double power-laws (not shown)."**

**Section 5:**

**Line 402 the vertical wind field .**

Line 427: We rephrased this sentence**: "This arises the question whether or not the elliptic clouds undergo predominant orientations." → "The tendency of elliptic cloud shapes raises the question whether or not such clouds undergo predominant orientations."**

Line 447: changed to **"this means that clouds are more stretched […]"**

Line 457: We slightly rephrased our last statement of Sec. 5: "Our variability in wind and geometry responses are, in turn, notably smaller and cases from additional campaigns might be beneficial. They may further classify our tendencies of wind-induced cloud elongation" → "Our variability in wind and geometry responses is notably smaller and cases from additional campaigns might be beneficial. They may further clarify our findings of wind-induced cloud elongation."

**Conclusions:**

Due to our supplementary analysis of cloud size and elongation relation, we added the following to the third concluding bullet point (Line 503):

**"Such biases can increase with cloud enlarging as these clouds seem more elongated by the winds. However, larger datasets are required for its verification."**

**IV) Response to the comments of the editor after reviewing the revised manuscript**

Under the given circumstances, we expressively thank the editor for reviewing the revised manuscript. Please find below our response to the comments (in *italics*) from the associate handling editor. Adapted modifications in the manuscript are highlighted **bold.** The line numbers here also refer to the document indicating the tracked changes after the review process (7 March 2022).

Line 16: do you mean "or single grid points that are removed by the speckle filter" ?
**Response:** We changed the phrase as follows: **"The radar encounters difficulties to represent clouds shorter than 200 m as they are either unresolved at all or are incorrectly displayed as single grid points.**

Line 114: I would say "As soon as a radar signal exceeds the radar sensitivity, the ... " because "zero reflectivity" cannot be recorded due to noise.
**Response:** we changed the wording accordingly

Line 351: omit repeated Fig. 9 reference?
**Response:** we deleted the repeat in referencing Fig 9.

Line 368: Do you mean "to prevent the requirement of introducing a scale break when using a single distribution" ?
**Response:** We rephrased this as follows: **"Due to the complexity of scale break origin, some studies, e.g. van Laar et al. (2019), suggest to apply exponential power law fits (Ding et al. 2014). They yield single distributions and prevent the requirement of introducing a scale break."**

Line 434ff: "Referring..." Please simplify this sentence, e.g. split it into two.
**Response:** we simplified our statement by splitting it into two sentences.

Line 522: seem -> seem to be
**Response:** we changed the wording accordingly

Line 522: "for verifying this finding" ?
**Response:** we changed the wording accordingly